**Investigation**

# Sex-specific evolutionary programs shape recombination rate evolution in house mice

Lydia K. Wooldridge,[1] Micah Pietraho,[1] Peyton DiSiena,[1] Sam Littman,[1] Benjamin Clauss,[1] Beth L. Dumont ID [1,2,3,]*

[1]The Jackson Laboratory, 600 Main Street, Bar Harbor, ME 04609, United States
[2]Tufts University, Graduate School of Biomedical Sciences, 136 Harrison Avenue, Boston, MA 02111, United States
[3]University of Maine, Graduate School of Biomedical Science and Engineering, 5775 Stodder Hall, Rm 15A, Orono, ME 04469, United States

*Corresponding author: The Jackson Laboratory, 600 Main Street, Bar Harbor, ME 04609, United States. Email: beth.dumont@jax.org

Recombination rates vary across species, populations, and sexes. House mice (*Mus musculus*) present a particularly extreme example. Prior studies have established large differences in global recombination rates between *M. musculus* subspecies and inbred strains, with males exhibiting more extensive variation than females. The observation of sex-limited variation has prompted the hypothesis that male and female recombination rates may evolve by distinct evolutionary mechanisms in *M. musculus*. Here, we formally evaluate this hypothesis in a phylogenetic framework. We combine cytogenetic estimates of genomic crossover counts with published data to compile a large dataset of sex-specific crossover rate estimates totaling >6,000 single meiotic cells from 31 genetically diverse inbred mouse strains representing five *Mus* species and four *M. musculus* subspecies. We show that the phylogenetic distribution of male recombination rates is well predicted by the underlying *Mus* phylogeny (phylogenetic heritability, $H_P^2 = 0.82$), contrasting with the weaker phylogenetic signal observed in females ($H_P^2 = 0.24$). *M. m. musculus* males exhibit a marked increase in recombination rate compared to males from other *M. musculus* subspecies, prompting us to test explicit models of lineage-specific evolution. We uncover evidence for an adaptive increase in male recombination rate along the *M. m. musculus* subspecies lineage but find no support for a parallel increase in females. Taken together, our findings confirm the hypothesis that recombination rate evolution in house mice is governed by distinct sex-specific evolutionary regimes and motivate future efforts to ascertain the sex-specific selective pressures and sex-specific genetic architectures that underlie these observations.

Keywords: recombination rate; phylogenetic comparative methods; house mice; genetic conflict; sex dimorphism

## Introduction

The rate of recombination is a fundamental parameter in genetics and evolutionary biology. Recombination rates influence the evolutionary trajectories of new alleles within populations (Webster and Hurst 2012), govern the introduction of novel haplotypes into the gene pool, and play a central role in theories concerning the origin and maintenance of sexual reproduction (Otto and Lenormand 2002). Recombination also shapes various genomic characteristics, such as base composition (Duret and Arndt 2008) and levels of both DNA (Halldorsson et al. 2019; Palsson et al. 2025) and haplotype diversity (Pritchard and Przeworski 2001; Greenwood et al. 2004). It is also essential for ensuring the accuracy of meiosis: both excessively high and abnormally low recombination rates can lead to chromosome segregation errors, resulting in aneuploid gametes or early meiotic failure (Hassold and Hunt 2001).

A key principle of evolutionary biology is that functionally significant traits should be subject to stabilizing selection for a narrow range of optimal values. In perplexing contrast to this prediction, recombination rates display extensive variability across multiple evolutionary timescales (Coop and Przeworski 2007; Smukowski and Noor 2011; Peñalba and Wolf 2020). Across eukaryotes, global recombination rates span three orders of magnitude (Stapley et al. 2017). Even within mammals, genome-scale recombination rates range 10-fold (Dumont 2017a). Recombination rates also vary at the population level. For example, recombination rates in outbred mice (Dumont et al. 2009), humans (Cheng et al. 2009; Fledel-Alon et al. 2011), sheep (Johnston et al. 2016), and *Drosophila* (Hunter et al. 2016) differ by a factor of at least two.

In many species, recombination rates also exhibit striking sex dimorphism, or heterochiasmy. For example, in mammals, females typically have higher global recombination rates than males, but males exhibit localized enrichment of recombination in telomeres (Broman et al. 1998; Sardell and Kirkpatrick 2020). While prior work has mapped loci with sex-limited or sex-specific effects on recombination rate in diverse vertebrate species (Kong et al. 2008, 2014; Liu et al. 2014; Ma et al. 2015; Halldorsson et al. 2019; Brekke et al. 2023; McAuley et al. 2024), the ultimate evolutionary causes of this sex dimorphism are largely unknown. The meiotic cells of the ovary and testis provide distinct, sex-specific milieus for recombination to unfold, establishing numerous plausible cellular mechanisms for the emergence of sex differences in recombination. Indeed, sex differences in the temporal progression of meiosis (Hunt and Hassold 2002), meiotic chromatin structure (Gruhn et al. 2013), meiotic gene regulatory programs (Turner 2007), and meiotic checkpoint stringency (Morelli and Cohen 2005) could establish sex-specific evolutionary pressures

on meiosis and recombination. Several theoretical hypotheses have also been advanced to explain heterochiasmy, including sex differences in the strength of epistasis (Lenormand 2003; Sardell and Kirkpatrick 2020), haploid selection (Lenormand 2003), meiotic drive (Brandvain and Coop 2012), and selection against gametic aneuploidy (Sardell and Kirkpatrick 2020). However, these theoretical hypotheses all presuppose that sex differences in recombination are adaptive (Sardell and Kirkpatrick 2020)—an assumption that has yet to be broadly established (Ritz et al. 2017).

We and others have previously suggested that house mice (*M. musculus*) present an especially valuable model system for studying the causes of sex differences in recombination (Liu et al. 2014; Dumont 2017b; Peterson and Payseur 2021). Using both cytogenetic and genetic approaches for quantifying genome-scale recombination rates, prior studies have shown that recombination rates are broadly conserved across female house mice, but more highly diverged across males of different house mouse strains and subspecies (Dumont and Payseur 2011a; Peterson and Payseur 2021). Moreover, several inbred mouse strains have been identified that show a reversal in the most common direction of sexual dimorphism for recombination rate (Dumont and Payseur 2011a; Peterson and Payseur 2021). For instance, males from several inbred strains of *M. m. musculus* have higher recombination rates than *M. m. musculus* females, contrasting the dominant trend of higher female recombination rates in strains from *M. m. domesticus* and *M. m. castaneus*. These directional shifts in heterochiasmy and the magnitude of male recombination rate divergence in house mice are surprisingly stark given that *M. musculus* subspecies diverged only 350–500 thousand years ago (Phifer-Rixey et al. 2020) and share a conserved karyotype.

While the cause(s) of these subspecies differences in recombination rate dimorphism is unknown, these observations would seem to suggest that male and female house mice evolve under different selective pressures or evolutionary regimes. Here, we set out to directly test this hypothesis using sex-specific phylogenetic modeling. We compiled a set of cytogenetic recombination rate estimates for 31 inbred mouse strains that capture each of the core *M. musculus* subspecies and four outgroup taxa. We then use whole genome sequences to reconstruct phylogenetic trees reflecting strain relationships and apply phylogenetic comparative methods to evaluate different modalities of sex-specific recombination rate evolution. Our findings reveal distinct evolutionary processes shaping male and female recombination rates and suggest that adaptive mechanisms have contributed to patterns of male recombination rate divergence in *M. musculus*.

## Methods
### Animal husbandry and experimental crosses
Wild-derived inbred strains CAROLI/EiJ, CZECHII/EiJ, GAIB/NachJ, JF1/MsJ, LEWES/EiJ, MANB/NachJ, MOLF/EiJ, PAHARI/EiJ, PERC/EiJ, PWK/PhJ, SARA/NachJ, SARB/NachJ, and SPRET/EiJ were obtained from The Jackson Laboratory (JAX) Repository and housed in a low barrier room within the JAX Research Animal Facility. Strain GOR/TUA (RBRC01242) was cryorecovered from frozen embryos purchased from the RIKEN BioResources Center and maintained by sib × sib mating. Mice from inbred strain POHN/DehMmJax were originally obtained from Dr. David Harrison's Laboratory at JAX and transferred to the Dumont Lab's private colony.

All mice were housed and handled in strict accordance with an animal care and use protocol approved by the JAX Animal Care and Use Committee (Protocol # 17021). All mice were provided with access to food and water ad libitum. Sexually mature males were euthanized by exposure to $CO_2$ at 8–26 weeks of age. Pregnant females were euthanized by $CO_2$ inhalation at 16.5–18.5 d post coitus. Pups were then dissected out of the pregnant female uterus and sacrificed by decapitation with sharp blades.

### Assaying autosomal crossover rate via cytogenetic analysis of MLH1 foci in pachytene cells
The localization of MLH1 on meiotic chromosome axes during prophase I approximates the frequency and distribution of crossovers and can be used as a proxy for the genomic crossover distribution and rate (Anderson et al. 1999). Meiotic cell spreads were prepared from adult testis tissue and fetal ovarian tissue as previously described (Peters et al. 1997) and immunostained following published protocols (Anderson et al. 1999; Murdoch et al. 2010). Slides were blocked in blocking media [1% donkey serum, 3% bovine serum albumin (300 mg/10 mL; Fraction V; Fisher Scientific), and 0.0005% Triton X-100 v/v in 1× PBS (pH = 7.4)] and antibodies were diluted in antibody dilution buffer (ADB; 1% normal donkey serum, 0.3% bovine serum albumin). The following primary antibodies were used at the specified dilutions: 1:75 mouse anti-MLH1 (BD, cat# 550838), 1:300 rabbit anti-SCP3 (Novus Biologicals, cat # NB300-231), and 1:100 human anti-centromere polyclonal (Antibodies, Inc., cat # 15-234). The following secondary antibodies were used at 1:200 dilution: donkey anti-mouse Alexa Fluor 488, donkey anti-goat Alexa Fluor 594, and donkey anti-human Coumarin AMCA (Jackson Immunoresearch). Slides were mounted in ProLongGold antifade (Promega) and imaged at 63× on a Leica DM6B upright epifluorescent microscope equipped with DAPI, GFP, and Texas Red fluorescent filters and a cooled monochrome 2.8-megapixel digital camera. Images were post-processed and analyzed in the Fiji software package (Schindelin et al. 2012).

A minimum of 40 late pachytene cells characterized by (i) the complete merger of SYCP3 signals from all autosomal homologs; (ii) a full complement of chromosomes; (iii) low background fluorescence; and (iv) bright, punctuate MLH1 signals were imaged for each inbred strain. Cells that were damaged during preparation or displayed bulbous chromosome termini (indicative of transition into diplotene) were not imaged. For each cell meeting these criteria, the total number of autosomal MLH1 foci was recorded. MLH1 foci on the sex chromosome bivalent were excluded, as the meiotic dynamics of the XY sex chromosomes are temporally decoupled from those of the autosomes (Kauppi et al. 2011; Acquaviva et al. 2020). In total, MLH1 foci counts were obtained for 1,266 cells across 57 individuals representing 15 strains. These totals include 15 females from five strains (GOR/TuA, PERC/EiJ, POHN/DehMmJax, PWK/PhJ, and SARB/NachJ), and 42 males representing 14 strains (Supplementary Table 1).

### Compilation of legacy MLH1 foci count data
We combined our newly collected MLH1 foci counts with previously published MLH1 data for genetically diverse inbred and out-bred male and female mice (Lynn et al. 2002; Dumont and Payseur 2011b; Peterson and Payseur 2021) (Table 1; Supplementary Table 1). We limit our focus to wild-derived inbred strains and outbred stocks, as the origin history of classical inbred mouse strains is complex and laboratory strains are unnatural hybrids carrying genomic ancestry from each of the three cardinal house mouse subspecies (*M. m. domesticus*, *M. m. castaneus*, and *M. m. musculus*) (Beck et al. 2000; Yang et al. 2007).

**Table 1.** Sex-specific MLH1 foci counts in genetically diverse *Mus* strains.

| Species | Subspecies | Strain | Male Avg MLH1 Count | N | SD | Female Avg MLH1 Count | N | SD | Reference |
|---|---|---|---|---|---|---|---|---|---|
| *M. caroli* | | CAROLI/EiJ | 29.67 | 58 | 2.56 | | | | This Study |
| *M. caroli* | | CAR/Rbrc | 27.02 | 57 | 2.99 | | | | (Peterson and Payseur 2021) |
| *M. musculus* | castaneus | CAST/EiJ | 21.96 | 132 | 1.85 | 25.92 | 13 | 3.33 | (Lynn et al. 2002; Dumont and Payseur 2011b; Peterson and Payseur 2021) |
| | castaneus | CIM | 22.67 | 109 | 2.19 | | | | (Dumont and Payseur 2011b) |
| | castaneus | HMI | 24.02 | 54 | 2.81 | | | | (Peterson and Payseur 2021) |
| | castaneus | POHN/DehMmJax | 22.75 | 67 | 2.20 | 23.37 | 59 | 2.80 | This study |
| | **castaneus** | **Subspecies Average**[a] | **22.85** | **4** | **0.86** | **24.65** | **2** | **1.80** | |
| *M. musculus* | domesticus | GAIB/NachJ | 21.00 | 52 | 1.73 | | | | This study |
| | domesticus | Gough | 23.39 | 417 | 2.65 | 28.03 | 335 | 4.23 | (Dumont and Payseur 2011b; Peterson and Payseur 2021) |
| | domesticus | LEWES/EiJ | 24.10 | 309 | 2.95 | 26.30 | 151 | 4.52 | This study; (Peterson and Payseur 2021) |
| | domesticus | MANB/NachJ | 24.18 | 59 | 2.30 | | | | This study |
| | domesticus | PERA/EiJ | 23.08 | 291 | 1.88 | | | | (Dumont and Payseur 2011b) |
| | domesticus | PERC/EiJ | 22.17 | 24 | 1.76 | 23.27 | 55 | 2.61 | This study; (Peterson and Payseur 2021) |
| | domesticus | SARA/NachJ | 24.06 | 88 | 2.42 | | | | This study |
| | domesticus | SARB/NachJ | 24.48 | 113 | 2.33 | 22.38 | 8 | 1.77 | This study |
| | domesticus | WSB/EiJ | 23.11 | 332 | 2.45 | 24.95 | 201 | 3.54 | (Dumont and Payseur 2011b; Peterson and Payseur 2021) |
| | **domesticus** | **Subspecies Average**[a] | **23.28** | **9** | **1.12** | **24.99** | **5** | **2.28** | |
| *M. musculus* | molossinus | JF1/MsJ | 25.64 | 45 | 1.92 | | | | This study |
| | molossinus | MOLF/EiJ | 23.68 | 180 | 2.41 | 27.62 | 21 | 4.24 | This study; (Peterson and Payseur 2021) |
| | molossinus | MSM/MsJ | 30.51 | 228 | 3.30 | 28.11 | 303 | 4.38 | (Peterson and Payseur 2021) |
| | **molossinus** | **Subspecies Average**[a] | **26.61** | **3** | **3.52** | **27.86** | **2** | **0.35** | |
| *M. musculus* | musculus | AST/Tua | 24.35 | 65 | 2.62 | | | | (Peterson and Payseur 2021) |
| | musculus | CZECHI/EiJ | 26.80 | 156 | 2.25 | | | | (Dumont and Payseur 2011b) |
| | musculus | CZECHII/EiJ | 22.41 | 66 | 2.29 | | | | (Peterson and Payseur 2021) |
| | musculus | GOR/TUA | 29.61 | 74 | 1.92 | 24.78 | 50 | 2.97 | This study |
| | musculus | KAZ/TUA | 23.18 | 254 | 2.90 | 25.71 | 182 | 3.93 | (Peterson and Payseur 2021) |
| | musculus | PWD/PhJ | 29.35 | 353 | 2.87 | 25.89 | 226 | 3.75 | (Dumont and Payseur 2011b; Peterson and Payseur 2021) |
| | musculus | PWK/PhJ | 31.41 | 44 | 1.90 | 25.89 | 98 | 3.32 | This study |
| | musculus | SKIVE/EiJ | 26.16 | 211 | 2.72 | 25.40 | 50 | 3.10 | (Peterson and Payseur 2021) |
| | musculus | TOM/TUA | 24.44 | 9 | 3.09 | | | | (Peterson and Payseur 2021) |
| | **musculus** | **Subspecies Average**[a] | **26.41** | **9** | **3.14** | **25.54** | **5** | **0.47** | |
| *M. musculus* | | **Species Average**[b] | **24.74** | **25** | **2.77** | **25.55** | **14** | **1.73** | |
| *M. pahari* | | PAHARI/EiJ | 25.78 | 40 | 1.54 | | | | This study |
| *M. spiceligus* | | PANCEVO/EiJ | 24.68 | 99 | 2.06 | | | | This study; (Peterson and Payseur 2021) |
| *M. spiceligus* | | SPI/TUA | 25.71 | 136 | 2.79 | 27.96 | 108 | 4.45 | (Peterson and Payseur 2021) |
| *M. spretus* | | SPRET/EiJ | 24.81 | 194 | 2.36 | 27.14 | 7 | 6.09 | This study; (Peterson and Payseur 2021) |

[a]For each subspecies, the mean MLH1 foci count was computed as the average of strain-level means; standard deviation was calculated across those strain-level means.
[b]The species-level average is the mean of average MLH1 foci counts across *M. musculus* strains; standard deviation was calculated across strain-level means.

Cells with <19 or >44 MLH1 foci were excluded on the basis that (i) a minimum of one crossover per chromosome pair is required for accurate meiotic segregation (Page and Hawley 2003) and (ii) cells with extreme numbers of MLH1 foci could represent staining artifacts. Within the Peterson and Payseur dataset, we further excluded cells assigned quality scores >4 to retain only the subset of high-quality data (see Peterson and Payseur 2021 for a description of quality scores). Overall, the combined dataset includes MLH1 foci counts from 6,277 pachytene-stage cells from 31 strain backgrounds spanning five *Mus* species. The species *M. musculus* is represented by multiple inbred strains from each of the three primary *M. musculus* subspecies (*M. m. domesticus*, *M. m. musculus*, *M. m. castaneus*), as well as multiple inbred strains of *M. m. molossinus*, a natural hybrid between *M. m. musculus* and *M. m. castaneus* (Yonekawa et al. 1988). Supplementary Fig. 1 presents a map showing the geographic origins of strains included in this dataset.

Data for several strains were collected by different investigators as part of independent studies. In these circumstances, we confirmed consistency across scorers using Mann–Whitney *U* tests. While some strains yielded significantly different MLH1 foci counts between studies, differences were numerically small in magnitude and likely reflect study differences in treatment of MLH1 foci on the sex chromosomes or strain drift (<1.2 foci or <5.5%; Supplementary Fig. 2, Supplementary Table 2).

## Genome sequencing and genomic data analysis

Publicly available whole genome sequencing data are available for 18 of the 31 inbred strains included in our MLH1 foci dataset (Supplementary Table 3). Three additional inbred strains (GAIB/NachJ, GOR/TUA, and POHN/DehMmJax) were *de novo* whole genome sequenced to ~30× coverage using paired end 150 bp Illumina sequencing (GOR/TUA and POHN/DehMmJax) or ~10× coverage using PacBio HiFi (GAIB/NachJ). For 8 of the remaining 10 strains with no corresponding whole genome sequence, we mined public sequencing archives for a whole genome sequence of a wild-caught mouse sampled in close geographic proximity to the inbred strain founders. Accession numbers for these wild-caught proxies are provided in Supplementary Table 3. We were unable to identify suitable sequences for two strains (CAR/TUA and SPI/TUA); these strains were excluded from the genomic and phylogenetic analyses described below.

The compiled genome sequences include a mixture of Illumina paired end, BGI-SEQ, and PacBio HiFi sequencing reads. In all cases, fastq files were accessed from their respective sources and subjected to adaptor read trimming, file splitting, and quality control with fastp (v. 0.23.4) (Chen et al. 2018). Reads were subsequently mapped to the GRCm39 reference mouse genome with bwa (v. 0.7.18; Illumina, BGI-SEQ) (Li and Durbin 2009) or pbmm2 (v. 2_1.16; HiFi). File merging and indexing were performed using SAMtools (v. 1.21) (Li et al. 2009), and duplicate reads were marked using SAMBLASTER (v 0.1.24) (Faust and Hall 2014). Bam files for samples sequenced across multiple libraries (*Mus pahari*/EiJ and CAROLI/EiJ) were subsequently merged using SAMtools (v. 1.21). Per sample variant calling was performed using DeepVariant in WGS mode (v. 1.6.1; Illumina, BGI-SEQ) or HiFi mode (PacBio HiFi samples) (Poplin et al. 2018). Per sample gVCF files were then merged using GLnexus (v. 1.4.1) under the DeepVariantWGS configuration to produce a joint call set (Yun et al. 2021). Sites were then filtered using BCFtools (v. 1.16) (Danecek et al. 2021) to include only autosomal biallelic single nucleotide variants with ≤10% missing data and genotype quality ≥30. All bioinformatic analyses were performed using containerized software on the The Jackson Laboratory's high performance compute cluster.

## Phylogenetic tree construction

Phylogenetic trees were constructed from the filtered joint call set. Briefly, variants were greedily thinned based on linkage disequilibrium (LD) to include only those sites with $r^2 < 0.2$ using PLINK (v. 2.00a2.3LM) (Purcell et al. 2007). This step reduced the VCF file from 152.85 million to 21.20 million SNPs and eliminated SNPs in perfect or high LD, maximizing the informativeness of remaining sites and reducing the computational burden of phylogenetic inference. This LD-thinned VCF file was then converted into FASTA format using a custom perl script (vcf_to_fasta_justSNPs.pl). Subsequent conversions from FASTA to Stockholm and Phylip alignment formats were made using the SeqIO module for BioPython (v. 3.12.9).

Quicktree (v. 2.0) was used to build a neighbor joining tree from the LD-thinned SNPs (Howe et al. 2002). We used the same SNP dataset to infer a maximum likelihood phylogenetic tree using phyml (v. 3.3) (Guindon et al. 2010). We specified a GTR model of nucleotide evolution, computed nucleotide frequencies from the empirical data, and estimated the transition/transversion ratio, proportion of invariant sites, and gamma distribution of rate classes via maximum likelihood. Executed code and Newick format trees are provided in Supplementary File 1.

Both neighbor joining and ML trees were rooted to the *M. pahari* strain *Mus pahari*/EiJ and rendered as ultrametric using the *chronos* command in the R package ape (Paradis et al. 2004; Paradis and Schliep 2019). Phylogenetic correlation matrices were constructed for each tree using the *vcv* command in ape, specifying corr = TRUE.

## Generalized linear mixed modeling and phylogenetic heritability estimation

The number of MLH1 foci on a single chromosome can be envisioned as a Poisson distributed variable constrained by the biological requirement for ≥1 crossover. Assuming crossovers accumulate independently on different chromosomes at the same underlying rate, the genome-wide MLH1 foci count can then be modelled as the sum of multiple Poisson random variables, one for each chromosome. While the sum of multiple Poisson variables is itself a Poisson distributed variable, this value can also be approximated by a Gaussian (normal) distribution by the Central Limit Theorem. To evaluate the appropriateness of modeling genome-wide MLH1 foci counts using a Gaussian distribution, we simulated random draws from a normal distribution with mean and variance equal to values calculated from our compiled dataset of male or female MLH1 foci counts (Supplementary Table 1). The resulting simulated data where then truncated to account for the fundamental meiotic requirement of one crossover per chromosome (males: ≥19; females: ≥20). In males, crossing over on the sex chromosomes is temporally delayed compared to the autosomes (Kauppi et al. 2011). As a result, an MLH1 focus is not consistently observed on the sex chromosomes in pachytene spermatocyte spreads, motivating our adoption of different biological lower bounds in males and females. Overall, MLH1 counts in both sexes are closely approximated by their respective truncated normal distributions (Supplementary Fig. 3). In contrast, a truncated Poisson distribution provides a visibly poorer fit to the observed data due to the underdispersion of observed counts (Supplementary Fig. 3).

For a neutrally evolving trait, the phenotypic divergence between taxa should be proportional to their genetic divergence. In this circumstance, the underlying phylogenetic tree relating samples ought to be a good predictor of the distribution of trait values across taxa. To evaluate this hypothesis, we applied a Bayesian generalized modeling approach to estimate the proportion of variance in sex-specific recombination rates that is attributable to the underlying strain phylogeny (Lynch 1991). Specifically, let $y_{i,j,k}$ be the MLH1 foci count for cell $k$ from individual $j$ of strain $i$, from study $s$. We model $y_{i,j,k}$ as a linear combination of 3 random effects using a truncated normal distribution with a lower bound of 19 (male MLH1 counts) or 20 (female MLH1 counts) to reflect the sex-specific biological constraints on the minimum number of MLH1 foci:

$$y_{ijk} = \mu + u_i + v_{ij} + w_s + e_{ijk} \qquad (1)$$

where:

- $y_{ijk,male} \geq 19$ and $y_{ijk,female} \geq 20$;
- $\mu$ is the phenotypic component shared by all members of the phylogeny;
- $u_i \sim N(0, \sigma^2_{Strain}\mathbf{A})$ is the random effect for strain $i$, with variance modelled according to phylogenetic correlation matrix $\mathbf{A}$;
- $v_{ij} \sim N(0, \sigma^2_{Individual})$ is the random effect for individual $j$ nested within strain $i$;

- $w_s \sim N(0, \sigma^2_{Study})$ is the random effect for study $s$;
- $e_{ijk} \sim N(0, \sigma^2)$ is a residual error term that captures variation due to measurement error, model misspecification, phylogenetic uncertainty, phenotypic plasticity, rapid evolution along terminal branches, and fluctuating selection.

Model parameters were estimated in a Bayesian hierarchical framework implemented using the brms R package (Bürkner 2017). Parameter estimates were obtained from 10,000 iterations of 4 replicate MCMC chains, with the No-U-Turn Sampler algorithm used to sample from the posterior distribution. Sampling was performed at every 10th iteration, with the first 50% of iterations discarded as burn-in. Priors were set for all model parameters as follows:

$$Model\ Intercept\ (\mu) \sim N(25, 3)$$

$$standard\ deviation\ of\ residuals \sim Student\ t(3, 0, 3)$$

$$standard\ deviation\ of\ random\ effects \sim Student\ t(3, 0, 5)$$

To assess model convergence, we first visually inspected trace plots for all variance components and confirmed that chains mixed well and showed no evidence of autocorrelation or drift (Supplementary Fig. 4). Second, we confirmed that the Gelman-Rubin diagnostic, $\hat{R}$, was 1.00 for all parameter estimates, indicating the convergence and thorough mixing of replicate MCMC chains. Third, we ensured that effective sample sizes for all model parameters were sufficiently large (>1,700 in all cases), indicating deep sampling from the posterior distribution. Lastly, we invoked the *pp_check* function in the bayesplot R package to visually confirm that simulated datasets from the posterior predictive distribution showed good agreement with the observed distribution of MLH1 foci counts (Supplementary Fig. 5; Gabry et al. 2019).

The estimated variances associated with model terms ($\sigma^2_{Strain}$, $\sigma^2_{Individual}$, $\sigma^2_{Study}$, and $\sigma^2_e$) were combined to calculate the phylogenetic heritability ($H^2_P$) of both male and female recombination rates:

$$H^2_P = \frac{V_{Strain}}{V_{Strain} + V_{Individual} + V_{Study} + V_e} \quad (2)$$

where $V_{Strain}$, $V_{Individual}$, and $V_{Study}$ are the estimated variances for the random strain, individual, and study effects, respectively. $V_e$ is the residual model error. Note that $H^2_P$ is identical to the intraclass correlation for the random strain effect. As the posterior distribution of $H^2_P$ values was not symmetrical (Supplementary Fig. 6), we used the posterior median as a point estimate of $H^2_P$. We present the 95% Highest Posterior Density (HPD) intervals to indicate the range within which the true parameter value lies, given the data and model assumptions.

The proportion of variance in MLH1 foci counts attributable to the random study and individual effects was calculated via the intraclass correlation method, analogous to equation (2).

### Modeling lineage-specific evolution
To explicitly model lineage-specific adaptive shifts in recombination rate, we used the Ornstein–Uhlenbeck (OU) phylogenetic modeling framework implemented in the R package ouch (Hansen 1997; Butler and King 2004; Cressler et al. 2015). OU models extend a neutral phylogenetic model of trait evolution (i.e. Brownian motion) by specifying the intensity of selection toward an optimal trait value along predefined lineages. Instead of trait values drifting randomly across a phylogeny, an OU model imagines traits being gently pulled toward a preferred state, much like a "magnetic" force. This magnetic pull is defined by three key parameters: the optimal trait value(s) favored by evolution (Θ), the strength of selection toward that optimum (α), and the amount of random variation around the optimum (σ²). The OU framework provides flexibility to distinguish between distinct evolutionary models. Certain branches can be user-specified as "foreground" lineages, which are hypothesized to evolve toward one optimum, and the remaining "background" lineages, which evolve toward a second trait value. For both male and female recombination rates, we fit a model with a unique trait optimum along the *M. m. musculus* lineage compared to the rest of the phylogeny. This model was compared to two null models: strict Brownian motion and a model with a single trait optimum shared across all lineages. Model parameters were estimated via unconstrained numerical optimization to maximize the likelihood, with model selection performed through comparison of corrected Aikake Information Criterion (AICc) values. Specifically, the model with the lowest AICc was interpreted as providing the best fit to the data.

## Results
### Species, subspecies, strain, and sex diversity in global recombination rates
We quantified global crossover rates via cytological evaluation of MLH1 foci in 1,271 pachytene stage cells from 57 animals representing 15 genetically diverse inbred mouse strains. We combined these data with previously published MLH1 foci counts from house mice (see Methods) to assemble a large dataset of genomic crossover rate estimates from over 6,000 pachytene cells from 31 mouse strains representing five independent *Mus* species and four *M. musculus* subspecies (Supplementary Table 1; Table 1). This dataset includes MLH1 foci counts derived from both spermatoctyes (males) and oocytes (females) for 15 inbred mouse strains, providing a resource for explicit testing of hypotheses of sex-specific recombination rate evolution.

Species in our dataset span ~5–7.5 million years of evolutionary divergence (Chevret et al. 2005; Thybert et al. 2018). Curiously, strains with the largest difference in MLH1 foci counts are not necessarily the most evolutionarily divergent. Across males, average MLH1 foci counts per strain range from 21.00 in the *M. m. domesticus* strain GAIB/NachJ to 31.41 in the *M. m. musculus* strain PWK/PhJ—two taxa that diverged <0.5 MYA (Table 1) (Phifer-Rixey et al. 2020). Considering only females, MLH1 foci counts range from 22.40 in SARA/NachJ to 28.14 in MSM/MsJ, strains that again derive from two closely related *M. musculus* subspecies (Table 1). With the exception of *M. pahari*, which has a diploid chromosome number of 48, all *Mus* species in our dataset are represented by a conserved 2N = 40 karyotype comprised of acrocentric chromosomes. Thus, the absence of a clear species-level effect on recombination rate divergence is not simply a consequence of evolutionary shifts in the chromosomal constraints on recombination (Dumont 2017a).

Prior work has established significant subspecies-level recombination rate divergence among males of the *M. musculus* species complex, with *M. m. musculus* males exhibiting markedly increased MLH1 foci counts compared to males from *M. m. domesticus* and *M. m. castaneus* (Dumont and Payseur 2011b; Peterson and Payseur 2021). The addition of data from several new *M. musculus* strains lends further support for these trends

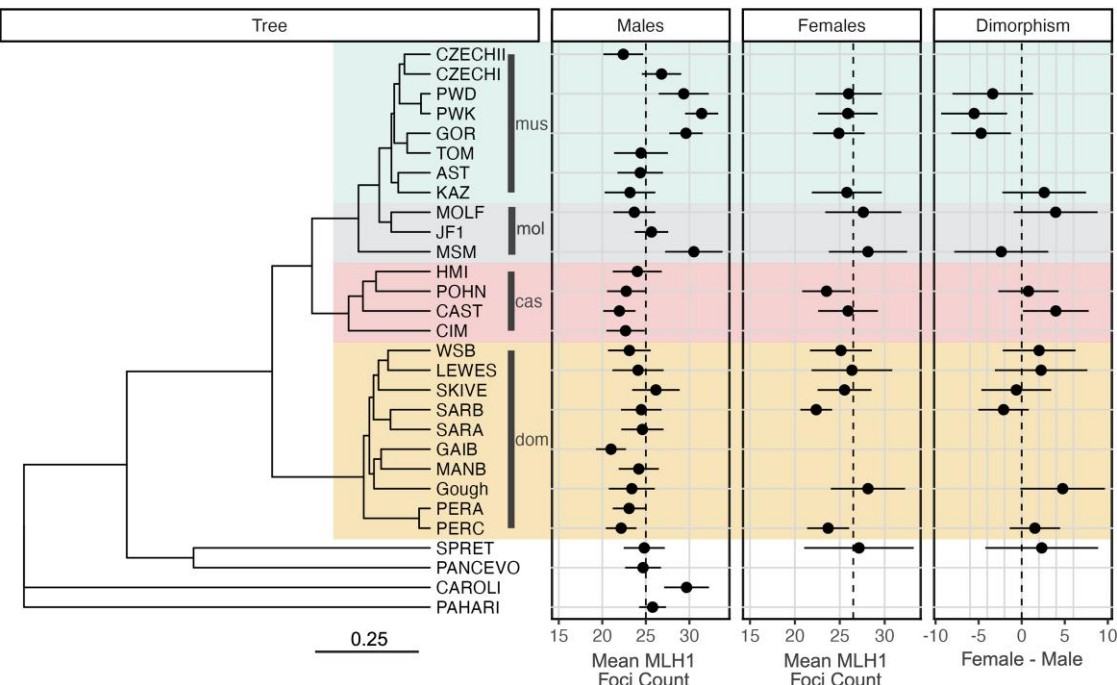

**Fig. 1.** Maximum likelihood phylogenetic tree for genetically diverse inbred mouse strains. Tree is rooted to the outgroup species *M. pahari* represented by inbred strain PAHARI/EiJ. For brevity, strain labels at the tips of the tree exclude laboratory codes. Wild-derived inbred strains from *M. musculus* subspecies are labeled and color-coded in gold (*domesticus*), red (*castaneus*), gray (*molossinus*), and turquoise (*musculus*). Strain SPRET is a wild-derived inbred strain of *M. spretus* origin, CAROLI is a representative of the *M. caroli* species, and PANCEVO is a wild-derived inbred strain of *M. spiceligus*. Panels to the right of the tree indicate average MLH1 foci counts (±1 standard deviation) for males and females for each strain, as well as the difference between male and female MLH1 foci counts (±1 standard deviation). Note that MLH1 foci counts are not available for females from all strains. Supplementary Fig. 7 presents a conceptually identical figure, but with a neighbor joining tree instead of a ML tree. Figure made using the ggtree package for R with aesthetic modifications in BioRender (Dumont, B, (2025) https://BioRender.com/45zmqzd).

(Fig. 1; Table 1). On average, males from inbred *M. m. musculus* strains have higher MLH1 foci counts than males from either *M. m. domesticus* or *M. m. castaneus* (*musculus*: 26.41; *domesticus*: 23.28; *castaneus*: 22.85; Table 1). Males from inbred strains of *M. m. molossinus* have similar MLH1 foci counts as *M. m. musculus* males (26.41 vs 26.61). *M. m. molossinus* is a natural hybrid between *M. m. musculus* and *M. m. castaneus* (Yonekawa et al. 1988), with genomic studies suggesting a dominant contribution of the *M. m. musculus* genetic background (Yang et al. 2007). This suggests the possibility of shared genetic factors in *M. m. molossinus* and *M. m. musculus* contributing to increased male recombination rates in these subspecies.

In contrast to the pronounced recombination rate divergence in *M. musculus* males, prior studies have revealed minimal divergence in recombination rate among females from the three cardinal house mouse subspecies (Peterson and Payseur 2021). This conclusion is reinforced by our addition of female MLH1 foci counts for several new inbred strains. The between strain variance in average MLH1 foci counts for *M. musculus* females is smaller than the corresponding measure for males (1.73 vs 2.77; Table 1), suggesting increased evolutionary constraint on female vs male recombination rates.

Overall, average female MLH1 foci counts are higher than the corresponding values for males (26.6 vs 25.0; Mann-Whitney U-test, $P = 3.92 \times 10^{-41}$), recapitulating the well-established pattern of heterochiasmy in house mice (Mallyon 1951; Dietrich et al. 1996; Shifman et al. 2006). However, at the level of individual inbred strains, there is considerable heterogeneity in the direction of this sex dimorphism. For example, male Gough mice have ~4.6 fewer MLH1 foci than their female counterparts, whereas male PWK/PhJ mice have on average 5.5 MLH1 foci more than females

(Table 1; Fig. 1). The majority of *M. m. musculus* strains exhibit increased male MLH1 foci counts compared to females. In contrast, most *M. m. domesticus* and *M. m. castaneus* strains show the opposite pattern of increased female MLH1 foci counts. It is notable that the one *M. m. musculus* strain defying this pattern (i.e. increased female MLH1 foci counts compared to males) derives from Kazakhstan, a location close to the presumed ancestral region where the *M. musculus* species first emerged (KAZ/TUA) (Boursot et al. 1993; Suzuki et al. 2013). This observation tentatively suggests that the directional change in heterochiasmy in *M. m. musculus* occurred after initial subspecies divergence and migration out of the ancestral cradle.

## Shared phylogenetic ancestry differentially accounts for variation in male and female MLH1 foci counts

We next set out to assess the phylogenetic distribution of MLH1 foci counts in male and female house mice using a modelling framework (see Methods). Specifically, we used a Bayesian linear mixed effects model to model sex-specific MLH1 foci counts as a function of three random effects (REs): a RE capturing variation across strains, a second RE to account for individual-level heterogeneity within strains, and the third RE to capture variation between studies represented in our MLH1 count dataset. To account for the nonindependence of phenotype measures across related strains, the strain level RE was structured by the phylogenetic correlation matrix computed from the strain phylogeny.

Under neutral phenotypic evolution, the distribution of MLH1 foci counts across strains should be well predicted by the structure of the underlying strain phylogeny. In contrast, if MLH1 foci counts are subject to strong stabilizing selection, evolve

**Table 2.** Estimated variance effects, regression coefficients, and intraclass correlations (ICC) from hierarchical Bayesian modeling.

| Dataset | Phylogeny | Lower bound | Random effects $\sigma_{Strain}$ | $\sigma_{Individual}$ | $\sigma_{Study}$ | Regression coefficients Intercept | $\sigma$ | Intraclass correlations Strain ($H_P^2$) (95% HPD) | Individual (95% HPD) | Study (95% HPD) |
|---|---|---|---|---|---|---|---|---|---|---|
| Males, 29 strains | ML | 19 | 6.59 | 1.29 | 1.05 | 25.3 | 2.59 | 0.8152 (0.59–0.90) | 0.0315 (0.02–0.06) | 0.0209 ($6.5 \times 10^{-5}$–0.27) |
| Females, 15 strains | ML | 20 | 3.26 | 1.96 | 2.54 | 24.46 | 4.51 | 0.2417 (0.05–0.54) | 0.0933 (0.03–0.17) | 0.1562 (0.003–0.61) |
| Females, 15 strains | ML | 19 | 2.66 | 1.63 | 2.26 | 25.04 | 4.14 | 0.2034 (0.04–0.49) | 0.0833 (0.03–0.15) | 0.1601 (0.003–0.65) |
| Males, 15 strains | ML | 19 | 6.16 | 1.45 | 0.97 | 24.77 | 2.7 | 0.7741 (0.51–0.89) | 0.0435 (0.02–0.08) | 0.0193 ($1.4 \times 10^{-5}$–0.32) |
| Males, 29 strains | NJ | 19 | 4.58 | 1.28 | 0.87 | 25.42 | 2.59 | 0.6891 (0.47–0.81) | 0.0547 (0.03–0.09) | 0.0251 ($4.2 \times 10^{-5}$–0.32) |
| Females, 15 strains | NJ | 20 | 2.69 | 1.94 | 2.74 | 24.39 | 4.52 | 0.1707 (0.04–0.43) | 0.0965 (0.04–0.17) | 0.1934 (0.005–0.64) |
| Females, 15 strains | NJ | 19 | 2.17 | 1.62 | 2.35 | 24.91 | 4.14 | 0.1466 (0.03–0.38) | 0.0875 (0.03–0.15) | 0.1840 (0.005–0.69) |
| Males, 15 strains | NJ | 19 | 4.62 | 1.45 | 0.9 | 24.68 | 2.7 | 0.6589 (0.40–0.83) | 0.0665 (0.03–0.12) | 0.0256 ($1.9 \times 10^{-5}$–0.37) |

exceptionally rapidly, or are highly sensitive to environmental differences, strain variation in this phenotype may not mirror strain ancestry. To assess these possibilities, we first quantified the phylogenetic heritability ($H_P^2$) of average MLH1 foci counts (Lynch 1991). The phylogenetic heritability is analogous to traditional broad sense heritability estimate: it represents the proportion of variation in a trait of interest that is explained by the underlying phylogeny relating the surveyed samples. We used the estimated variance components from our fitted models to derive sex-specific estimates of $H_P^2$. Male MLH1 foci counts exhibit high phylogenetic heritability across a maximum likelihood (ML) tree of *Mus* strains (median posterior $H_P^2 = 0.81$; 95% HPD interval: 0.59–0.90; Table 2). In contrast, $H_P^2$ for female MLH1 foci counts is markedly lower, indicating that shared evolutionary descent does not strongly predict the strain distribution of this trait (median posterior $H_P^2 = 0.24$; 95% HPD interval: 0.054–0.54; Table 2). For both sexes, the variation in MLH1 foci counts due to study-level differences and inter-individual variation is modest, accounting for <16% and <10% of the total variance, respectively (Table 2).

Recognizing that the discordant findings could reflect the increased number of strains with MLH1 foci count data for males compared to females, we pruned the ML tree and pared down our male MLH1 dataset to include only the 15 strains with MLH1 foci counts from both sexes (Table 1). We then re-estimated $H_P^2$ on the thinned male dataset. Our findings recapitulate those from the original, larger data (males: median posterior $H_P^2 = 0.77$, HPD interval: 0.51–0.89; Table 2). Similarly, differences in the lower bound specified in our sex-specific models do not meaningfully contribute to qualitative differences in $H_P^2$ between the sexes. Modeling female MLH1 foci counts with a lower bound of 19 (rather than 20), has little quantitative effect on variance component estimates or $H_P^2$ (Table 2).

We next assessed the robustness of our results to implicit assumptions of the Bayesian linear mixed effects models fit to estimate variance components. First, our models assume that MLH1 foci counts follow a truncated Gaussian distribution. We confirmed the appropriateness of this assumption via simulation, establishing that the observed distributions of male and female MLH1 foci counts are well-approximated by truncated normal distributions (Supplementary Fig. 3). Second, the Bayesian framework used to estimate model parameters assumes that posterior samples are representative of the true posterior distribution. We performed extensive diagnostic testing to confirm the convergence of sampling chains (Supplementary Fig. 4) and validate that draws from the posterior distribution approximate the observed sex-specific distributions of MLH1 foci counts (Supplementary Fig. 5). Third, phylogenetic heritability estimates

assume that the tree specified in the model is correct. We performed phylogenetic inference using whole genome sequence data from analyzed strains or their close geographic proxies from the wild. The reliance on genomic data effectively integrates over regional fluctuations in evolutionary history due to incomplete lineage sorting or introgression, ensuring that inferred trees are an accurate portrayal of species and strain relationships. Our trees also recapitulate known relationships among *Mus* species and cluster strains by subspecies and geographic origin (Fig. 1), as expected (Chevret et al. 2005). Further, our findings are robust to the method of phylogenetic inference, as we recover qualitatively similar $H_P^2$ estimates when accounting for ancestry using an NJ tree (males, median posterior $H_P^2 = 0.69$, 95% HPD interval: 0.47–0.81; females, median posterior $H_P^2 = 0.17$, 95% HPD interval: 0.04–0.43; Table 2; Supplementary Fig. 7). Lastly, the ML (or NJ) trees informing our $H_P^2$ estimates for male and female MLH1 foci count data are identical. Thus, quantitative sex differences in $H_P^2$ estimates cannot be explained by differences in the phylogeny.

Overall, our findings indicate that phylogenetic relatedness is a stronger predictor of male recombination rates than female recombination rates. This finding could be attributed to one or more explanations. First, males may harbor increased additive genetic variance for recombination rate compared to females (*i.e.* male recombination rates have higher genetic heritability). Second, our findings could imply that female recombination rates are subject to more intense stabilizing selection across the *Mus* taxa evaluated here, with reduced additive trait variance in MLH1 foci counts leading to moderated estimates of $H_P^2$. Third, female recombination rates may be more sensitive to environmental perturbations than male recombination rates. Finally, the absence of a strong phylogenetic signal in females could be due to extremely rapid recombination rate evolution in this sex, with trait values fluctuating so rapidly in time that they override signals of shared evolutionary history. We discount this final interpretation as unlikely given the lower between strain variance in MLH1 foci counts in females compared to males (Table 1).

## Modeling of sex-specific selection regimes for recombination rates in *M. musculus*

Having discovered distinct phylogenetic trends in recombination rate evolution between males and females, we next sought to explicitly model defined scenarios of sex-specific recombination rate evolution in a phylogenetic context. Given that *M. m. musculus* males consistently exhibit among the highest MLH1 foci counts and capture a shift in the direction of sex dimorphism for recombination rate (Fig. 1), we were particularly interested in evaluating the possibility of an adaptive increase in male recombination

**Table 3.** Evaluation of distinct phylogenetic models of sex-specific recombination rate evolution.

| Tree | Sex | Model | α | σ² | θ$_a$ | θ$_{Mmm}$ | Likelihood | df | AIC - corrected |
|------|-----|-------|---|-----|-----|-------|-----------|-----|-----------------|
| NJ | M | Brownian motion | | 13.37 | | | −50.80 | 2 | 106.23 |
| NJ | M | Single optimum | 2.63 | 30.66 | 24.21 | | −49.51 | 3 | 106.34 |
| NJ | M | *M. m. musculus* | 646.11 | 5607.01 | 23.31 | 26.44 | −47.31 | 4 | 104.98 |
| NJ | F | Brownian motion | | 6.67 | | | −24.51 | 2 | 54.36 |
| NJ | F | Single optimum | 9.90 | 46.72 | 25.39 | | −22.17 | 3 | 53.34 |
| NJ | F | *M. m. musculus* | 150.03 | 700.31 | 25.29 | 25.64 | −22.11 | 4 | 57.94 |
| ML | M | Brownian motion | | 35.69 | | | −54.97 | 2 | 114.57 |
| ML | M | Single optimum | 8.29 | 92.12 | 24.00 | | −49.44 | 3 | 106.21 |
| ML | M | *M. m. musculus* | 571.97 | 4943.69 | 23.31 | 26.44 | −47.31 | 4 | 104.98 |
| ML | F | Brownian motion | | 10.39 | | | −24.48 | 2 | 54.28 |
| ML | F | Single optimum | 10.52 | 50.01 | 25.37 | | −22.04 | 3 | 53.09 |
| ML | F | *M. m. musculus* | 10.75 | 51.05 | 25.35 | 25.49 | −22.04 | 4 | 57.80 |

α: strength of directional force toward optimal trait value, units are inverse of time.
σ²: magnitude of random perturbations tolerated about the optimal trait value, units measured as MLH1 foci × time.
θ$_a$: ML estimate of optimal MLH1 foci count across the tree.
θ$_{Mmm}$: ML estimate of optimal MLH1 foci count value along *M. m. musculus* lineage.

rates along the *M. m. musculus* subspecies lineage. We modeled this evolutionary scenario as an Ornstein–Uhlenbeck process, specifying a unique adaptive trait optimum and evolutionary rate along the *M. m. musculus* lineage relative to the rest of the phylogeny. We then compared this evolutionary model to null models wherein (i) the evolution of MLH1 foci counts across all strains is determined by Brownian motion or (ii) the evolution of MLH1 values is governed by selection toward a single trait optimum across the entire *M. musculus* phylogeny.

A model with a unique adaptive optimum in *M. m. musculus* is a better fit to the phylogenetic distribution of male MLH1 foci counts than a Brownian motion model or model with a single global trait optimum, as assessed by the corrected Aikake Information Criterion (AICc; Table 3). In contrast, both simplified models provide better fits to female MLH1 foci counts than the more complex model with a distinct adaptive optimum for *M. m. musculus*. We recover qualitatively identical findings when modeling these evolutionary regimes along both NJ and ML trees (Table 3). For both trees, the maximum likelihood estimate of the optimum male trait value for the *M. m. musculus* lineage (θ$_{,Mmm}$) is ~3 MLH1 foci higher than that for background lineages, and estimates of the strength of selection toward the *M. m. musculus* adaptive optimum (α) are two orders of magnitude greater than that for the single optimum model. Taken together, these findings indicate that global recombination rates are evolving according to different evolutionary regimes in *M. musculus* males and females, and provide evidence for an adaptive, lineage-specific increase in male recombination rates in *M. m. musculus*.

To further confirm these findings, we next modeled the magnitude of sex dimorphism in MLH1 foci counts between females and males according to the same evolutionary regimes. Although our dataset is modest in size (n = 15 taxa), model fitting with the ML tree favors an evolutionary regime in which a distinct trait optimum is favored in *M. m. musculus* compared to other lineages (AICc, musculus lineage = 69.06; AICc, single optimum = 69.28). We obtain qualitatively identical results when modeling these evolutionary scenarios along the NJ tree (AICc, musculus lineage = 69.06; AICc, single optimum = 70.02; Supplementary Table 4).

## Discussion

The rate of recombination shapes outcomes of evolutionary processes in nature, but we lack a comprehensive understanding of how the molecular phenotype of recombination itself evolves (Johnston 2024). Prior work has established significant variation in recombination rate between species, within populations, and between males and females, but few studies have analyzed this phenotypic diversity in a phylogenetic framework (Lenormand and Dutheil 2005; Cooney et al. 2021; Szasz-Green et al. 2025). Here, we combined MLH1 foci counts across 15 genetically diverse inbred mouse strains with legacy datasets to evaluate the evolution of recombination rates across 31 wild-derived inbred house mouse strains in a sex-specific phylogenetic context. Our data confirm prior speculation that recombination rate evolution has proceeded under different evolutionary regimes in male and female house mice (Dumont and Payseur 2011a; Peterson and Payseur 2021). We show that female recombination rates display only a weak phylogenetic signal in *Mus* and exhibit a general trend of evolutionary stasis. In contrast, the phylogenetic distribution of male recombination rates is well-approximated by the structure of the underlying phylogeny and has been influenced by a lineage-specific increase in *M. m. musculus*.

Curiously, elevated male recombination rates are observed in only a subset of our surveyed *M. m. musculus* strains (PWD/PhJ, PWK/PhJ, and GOR/TUA). Male recombination rates in other *M. m. musculus* strains (CZECHI/EiJ, CZECHII/EiJ, TOM/TUA, AST/TUA, and KAZ/TUA) are close to, or even less than, the phylogeny-wide mean (Fig. 1). Thus, the male-specific increase in recombination rate in *M. m. musculus* is not universal across this clade. These observations can be interpreted through the lens of mouse demographic history. Current models posit that the three core *M. musculus* subspecies emerged from a central population in southwestern Asia and radiated westward, northward, and eastward very recently and nearly simultaneously (<0.5 MYA), giving rise to the *domesticus*, *musculus*, and *castaneus* subspecies lineages, respectively (Boursot et al. 1993; Phifer-Rixey et al. 2020). As subspecies ranges expanded, they came into secondary contact, leaving footprints of introgression and hybridization in contemporary wild mouse genomes (Fujiwara et al. 2022; Morgan et al. 2022). Further, because of their recent evolutionary origins, many loci in mouse genomes exhibit incomplete lineage sorting due to ancestral allele sharing between subspecies (White et al. 2009; Keane et al. 2011). Two of the high recombination rate *M. m. musculus* strains (PWD/PhJ and PWK/PhJ) derive from wild-caught mice sampled from close to a natural hybrid zone between *M. m. domesticus* and *M. m. musculus* that transects Europe (Macholán et al. 2012), while the wild-caught progenitors of the third high recombination rate *M. m. musculus* strain (GOR/TUA) derive from a more distant location in Siberia (Gorno-Altaisk, Russia). Future work will be required to determine whether the

extremely high recombination rate phenotype in these three geographically isolated strains (i) reflects the population sharing of alleles that emerged uniquely in *M. m. musculus* following subspecies migration out of the ancestral region, (ii) is due to ancestral alleles that were subsequently lost in *M. m. castaneus* and *M. m. domesticus*, or (iii) is attributable to recent introgression events between *M. m. musculus* populations, potentially aided by human migration and the status of house mice as human commensals. It is also possible that high recombination rates in these strains reflect parallel adaptation to shared environmental pressures. Indeed, prior work has revealed that temperature can drive adaptive differences in recombination rates in *Drosophila* (Samuk et al. 2020).

Our findings raise a key, outstanding question: what are the biological mechanisms that drive the sex-specific evolution of recombination rate in house mice? On a genetic level, sex-specific recombination rates could be underlain by alleles with sex-specific, sex-limited, or sex-linked effects on recombination (Pennell and Morrow 2013; Mank 2017a, 2017b). Linkage studies have identified numerous recombination rate modifiers, including alleles with opposite effects on recombination in males and females (Kong et al. 2008; Halldorsson et al. 2019), alleles with effects in only one sex (Kong et al. 2014; Ma et al. 2015; Brekke et al. 2023; McAuley et al. 2024), and sex-linked recombination rate modifiers (Murdoch et al. 2010; Dumont and Payseur 2011a; Liu et al. 2014). Our on-going efforts to define the genetic architecture of sex-specific recombination rates in *M. musculus* stand to provide further insight into the molecular underpinnings of this dimorphic and labile trait.

The discovery of a sex- and lineage-specific increase in recombination rate in *M. musculus* is especially exciting in the context of prior work on the genetic control of recombination in this system. Multiple large- and moderate-effect modifiers of male recombination rate have been mapped to the X chromosome and autosomes in intersubspecific crosses between *M. m. musculus* and *M. m. castaneus*, with X-linked and autosomal modifiers exhibiting antagonistic effects (Dumont and Payseur 2011a; Liu et al. 2014; Dumont 2017b). Specifically, in males, X-linked alleles from the low recombination rate subspecies (*M. m. castaneus*) confer an increase in recombination rate, whereas autosomal alleles from *M. m. castaneus* are associated with recombination rate decreasing effects. One of these X-linked modifiers overlaps a locus implicated in hybrid male sterility in intersubspecific crosses between *M. m. musculus* and *M. m. domesticus*, suggesting a possible link between recombination and the emergence of nascent subspecies barriers (Balcova et al. 2016). More recent work has demonstrated that a genetic modifier of female recombination rates is also resident on the mouse X chromosome (Liu et al. 2014; Balcova et al. 2016; Dumont 2017b), although additional work is required to demonstrate that the loci identified in males and females colocalize.

From an evolutionary perspective, support for unique sex-specific evolutionary programs implies that natural selection favors distinct recombination rate optima in males and females. Although aberrant recombination profiles are frequently associated with aneuploidy or infertility (Hassold and Hunt 2001; Oliver et al. 2008; Lu et al. 2012; Bell et al. 2020; Carioscia et al. 2025), the relationship between natural variation for recombination rate and reproductive fitness remains poorly understood (Ritz et al. 2017). Natural variation for recombination rate is shaped by adaptation in wild *Drosophila* populations (Samuk et al. 2020), but the sex-limited nature of recombination in drosophilids and other differences in life history may limit the

transferability of this finding to mammals. Human females with higher recombination rates birth more children (Kong et al. 2004; Campbell et al. 2015), but it is not clear whether this correlation reflects the increased fitness of females with higher recombination rates or the increased odds a fertilized oocyte with elevated recombination rate survives early development in older mothers. A recent simulation study concluded that human recombination rates are constrained by fitness costs for excessively high and low recombination rates, and that the fitness costs associated with excessively high recombination rates may be higher in males than females (Drury et al. 2023). However, to our knowledge, there exists no empirical evidence that natural selection promotes sex-specific differences in recombination rates.

Unfortunately, evaluating the possibility that the sex-specific variation in MLH1 foci counts observed across *M. musculus* is directly linked to fitness is not currently possible as we lack estimates for fitness-associated traits for all but a few of the strains surveyed in our study. Colony breeding records document strain-specific breeding performance and may provide a read-out of strain fitness. However, the reproductive performance of inbred strains is an amalgamation of male and female factors that cannot be readily disentangled to permit sex-specific estimates of fitness. Further, the interpretation of fitness in inbred strains is potentially confounded by the inbreeding process itself, which is associated with the fixation of deleterious variants at a rate proportional to the mutation load of the original parent population. These complexities call attention to the need for studies that profile sex-specific recombination rates and fitness measures in natural or outbred populations.

Although we lack direct experimental evidence for sex differences in recombination rate fitness optima, our finding that the evolution of recombination in house mice has unfolded according to sex-specific evolutionary regimes is strongly consistent with this possibility. Sex differences in trait fitness optima can impose a genetic conflict if alleles that advance the trait value in the direction favoring one sex have a negative effect on the fitness of the other sex. Such conflict can be genetically resolved via intralocus mechanisms, including sex-limited gene expression (Mank 2017a), sex-biased epigenetic silencing (Pennell and Morrow 2013), dominance reversal (Barson et al. 2015), gene duplication (VanKuren and Long 2018), or sequestration of genes with sexually antagonistic fitness effects to the sex chromosomes (Roberts et al. 2009; Mullon et al. 2012). Alternatively, sexual conflicts can be resolved via interlocus mechanisms involving distinct loci with opposing fitness effects in the two sexes. This latter scenario can fuel an evolutionary arms race between the sexes, wherein male-beneficial (female-detrimental) alleles impose a selective advantage for the emergence of female-beneficial (male-detrimental) alleles elsewhere in the genome (Schenkel et al. 2018). Overall, genetic conflict can maintain exceptionally high levels of trait polymorphism, propel rapid trait divergence, and shape key aspects of trait architecture (Mank 2017b; Schenkel et al. 2018).

These telltale hallmarks of interlocus genetic conflict are evident in the (1) high levels of recombination rate polymorphism within house mouse subspecies (Dumont and Payseur 2011b; Peterson and Payseur 2021), (2) unique, sex-specific patterns of accelerated recombination rate divergence across *Mus* (this study), and (3) the genetic control of recombination by X-linked and autosomal genetic loci with antagonistic effects in *M. musculus* (Murdoch et al. 2010; Dumont and Payseur 2011a). Thus, our findings layer new evidence onto the hypothesis that recombination rate variation across house mice may be driven, at least in part,

by genetic conflict (Dumont 2017b). Additional work is clearly required to establish the validity of this hypothesis and identify the aspects of meiosis that may be subject to conflicting evolutionary pressures in males and females. Further, whether genetic conflict is a common mechanism shaping the evolution of recombination rates across taxa, or a potential phenomenon specific to house mice, warrants future experimental testing.

## Data availabilty statement

The MLH1 foci count data analyzed in this report are provided in Supplementary Table 1. Supplementary File 1 contains the Newick format phylogenetic trees used for phylogenetic analysis and model fitting. Raw fastq sequencing data used in phylogenic reconstruction are available at the accession numbers listed in Supplementary Table 3. Whole genome sequence data for the three strains newly sequenced as part of this study POHN/DehMmJax, GOR/TUA, and GAIB/NachJ are provided on the SRA Archive under PRJNA1276540. Code used to perform statistical analyses, model fitting, and generate figures can be found at https://doi.org/10.5281/zenodo.17180464.

Supplemental material available at GENETICS online.

## Acknowledgments

We gratefully acknowledge the Genome Technologies Scientific Services at The Jackson Laboratory for their contributions to this work. We also thank members of the Research IT team at The Jackson Laboratory for providing access and support to the high-performance computing resources that made this project possible. Lastly, we thank members of the Dumont Lab for helpful discussions and feedback.

## Funding

This work was supported by a CAREER award from the National Science Foundation to BLD (DEB1942620).

## Conflict of interest

None declared.

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

*Editor: T. Lenormand*