## [Peer Review File · Genetics]

Sex-Specific Evolutionary Programs Shape Recombination Rate Evolution in House Mice

Lydia Wooldridge, Micah Pietraho, Peyton DiSiena, Sam Littman, Benjamin Clauss, and Beth Dumont

NOTE: The reviews and decision letters are unedited and appear as submitted by the reviewers.

In extremely rare instances and as determined by a Senior Editor or the EIC, portions of a review may be redacted. If a review is signed, the reviewer has agreed to no longer remain anonymous.

The review history appears in chronological order.

Review Timeline:

Submission Date:	2025-06-15
Editorial Decision:	2025-08-17
Resubmission Received:	2025-09-23
Accepted:	2025-11-06

August 17, 2025

GENETICS-2025-308280

Recombination Rates Are Governed by Sex-Specific Evolutionary Programs in House Mice

Dear Dr. Dumont:

Three experts in the field have reviewed your manuscript, and I have read it as well. I agree that the findings are interesting and convincing. Documenting how heterochiasmy varies within and between closely related species is certainly useful to eventually help to understand its evolutionary origin. While your manuscript is not currently acceptable for publication in GENETICS, we would welcome a substantially revised manuscript. Reviewers have comments and concerns to be addressed in a revised manuscript. You can read their reviews at the end of this email. Their remarks and comments offer a clear path for this revision.

We look forward to receiving your revised manuscript. Please let the editorial office know approximately how long you expect to need for revisions.

Upon resubmission, please include:

1. A clean version of your manuscript;
2. A marked version of your manuscript in which you highlight significant revisions carried out in response to the major points raised by the editor/reviewers (track changes is acceptable if preferred);
3. A detailed response to the editor's/reviewers' feedback and to the concerns listed above. Please reference line numbers in this response to aid the editor and reviewers.

Your paper will likely be sent back out for review.

Additionally, please ensure that your resubmission is formatted for GENETICS

<https://academic.oup.com/genetics/pages/general-instructions>

Follow this link to submit the revised manuscript: Link Not Available

Sincerely,

Thomas Lenormand
Associate Editor
GENETICS

Approved by:
Anthony Long
Senior Editor
GENETICS

Reviewer #1 :

This paper by Wooldridge et al examines heterochiasmy in crossover counts from cytogenetic data across 31 inbred strains across different species and subspecies of the genus *Mus*. They integrate newly generated data with previous studies to amass MLH1 foci data for >6,000 meiotic cells. They use whole genome sequence data to build phylogenies (neighbour-joining and ML) with high confidence, allowing them to determine the phylogenetic signal of crossover count data within each sex. They show that male recombination rates have a substantial phylogenetic signal ($H^2 = 0.45$) compared to females ($H^2 = 0.01$). They propose that males follow an adaptive optimum along the lineage, whereas females have a global trait optimum across all strains.

Overall, I think the dataset is impressive, and the paper makes a valuable contribution to the literature by showing that heterochiasmy can be variable within and between closely related species, and adds to a growing body of evidence that female and male recombination are not directly equivalent. The paper is generally well-written and the analysis is (mostly) clear. Nevertheless, I have several comments/questions to be addressed before publication.

MAJOR COMMENTS

1. In lines 80-83 (and indeed throughout the paper), the authors should acknowledge more that heterochiasmy is frequently

underpinned by different genetic architectures in females and males. This has been repeatedly demonstrated in vertebrates (humans, cattle, pigs, deer, mice, sheep, goats, sparrows, salmonids) and often (but not always) implicates a suite of meiotic loci of modest to major effect to be involved (RNF212, RNF212B, PRDM9, SYCP2, MSH4, MEI1, etc). This is relevant for the current paper, as it provides a clear potential mechanism by which sex differences can rapidly evolve. It also provides a low-hanging fruit for future analyses of genomic data in this dataset by the authors to determine if there are phenotypic associations and/or signatures of selection.

2. I had several questions about the phylogenetic modelling:

- (a) It is not clear if the phenotypic value is a point estimate of the mean phenotypic value per strain, or if it incorporates all measures within each strain (possibly with the strain identity fit as a random effect to account for pseudoreplication).
- (b) Given that the authors acknowledge potential differences between data sources, it may have been appropriate to fit the data source as a random effect in the model (to account for between dataset variance).
- (c) To offer a potential solution if not possible in ape, I believe this is possible in MCMCglmm (Bayesian approach) that could allow better fitting and interpretation of the error around the estimates (which is never quite clear in the results of this manuscript).

3. I must admit that I did not fully understand what the modelling of lineage-specific adaptive shifts was doing, or what it means biologically (Lines 270-277). It is not clear what "the inclusion of additional trait parameters" of an optimal trait and selection intensity really means, or what these are, or how these are fit (this indicates a reproducibility issue - it's not enough to just signpost the code). What does a unique trait optimum along a lineage mean? What are background lineages discussed in lines 449-461? I apologise if my comment is vague, but I still didn't fully understand what had been done after reading the results and discussion. Please help the reader more (even an apparently experienced one such as myself).

4. There is substantial discussion of sexual conflict in the discussion (lines 563-584, also in lines 49-50), but I think it should be toned down, as it is not directly supported by findings of this manuscript. First, I don't agree that differences in phenotype or allelic effects inherently mean there is conflict, e.g., that an allele that advances a trait value in one sex will always have a negative effect in the other sex (as implied in lines 565-567). Second, and more generally, I think that there is no evidence of high trait polymorphism, an arms race, variation in trait architecture etc. in this study that supports this argument. It would be good to know *how* these hypotheses could be tested in future work.

5. There needs to be more considerations on how differences in the female sample size may affect the estimates (e.g., what is observed when males are down-sampled to reflect a similar dataset), and how much of the signal could be driven by a few members of *Mus musculus* exhibiting particularly high male values.

ABSTRACT and SUMMARY

The abstract could benefit from having the actual values for the phylogenetic signal.

Lines 33-36: On my first read, I did not really understand what this text was saying. I think that it could be rephrased in terms of biological meaning - the female part is quite clear but it was less clear what a "unique adaptive optimum" in the male lineage means.

METHODS

Line 172: Reproducibility: What scale is this quality score on, and how is this obtained?

General comment: Potential issues with karyotype variation are not discussed until quite late in the paper. A comment should be made earlier, and any species deviating from this should either be excluded or indicated in Tables and Figures.

Table 1: What does the bold text represent?

Figure S1: a small point, but this visualisation does not represent the underlying data well (sample sizes etc.) - it would be better to have transparent boxes with the data points jittered underneath.

Lines 181-186: I was a bit surprised that the authors did not deem differences in the inclusion of sex-chromosomes to be an issue, as this seems relevant to the study. See my point 2(b) above.

Lines 210-223: did the authors consider investigating potential effects of structural variants, or is this unlikely to be an issue?

Lines 225-243: what outgroups were selected, or is this not required?

RESULTS

Small comment: the paper reads like the results were supposed to be read before the methods, and also there is a bit of a

Results & Discussion vibrate in parts (e.g., lines 416-427). The paper was easy to read, so I don't mind it too much, but some information could have been useful earlier e.g., the karyotype information, divergence dates, known differences in mouse (consider whether Lines 312 - 324 could even be in the introduction).

Figure 1: This is a lovely figure but I have a few suggestions. The gridlines could match each branch rather than the species count as it would help direct matching of branch tips with the data point. The standard errors cannot be seen on most points - is this because they are so small? Seems odd based on the table information, but if so, could be good to indicate this fact in the legend. It may also be good to indicate the species of SPRET, PANCEVO, CAROLI in the legend.

Lines 342-344: how was the variance calculated - as the raw values or that estimated from the phylogenetic model? I also believe it is more appropriate to compare coefficients of variation rather than raw variance between different groups.

Lines 357-359: I agree with the word tentative here - I don't think there is enough evidence to back up this statement. Perhaps it is more appropriate to put this as a future hypothesis in the discussion.

Line 377: "not genetically determined" - I found this statement a bit odd. I get that this is a possibility, but I think there is a lot of evidence that there is likely substantial genetic determination, not only through major effect loci (see point 1 above), but also through heritable variation in chromatin conformation & accessibility.

Lines 380-386: in reporting these results, it would be appropriate to give more statistical parameters, such as the range and distribution of values obtained under the resampling regime.

Lines 418-419: I would argue that having more stringent genetic controls could lead to lower heritability as there would be less tolerance to variation in the trait? Similarly, in most other vertebrate systems where individual measures have been used, males often show lower phenotypic variance and lower heritability within populations. I'm not saying that this has to be the case in your data, but I think that it is worth considering.

DISCUSSION

Lines 477-479: There have been several studies that have done this between species that should probably be acknowledged, including Lenormand & Dutheil (2005, PLoS Genetics), Cooney et al (2021, Evolution), Kivioja & Rastas (2024, BioRxiv).

General comment: given the discussion of hybrid male sterility in paragraph 524-538, could PRDM9 play a role in heterochiasmy?

Reviewer #2 :

This research and manuscript is very inspiring since it is very difficult to study evolution of non-protein related processes. The proposed phylogenetic indicator or recombination rate is very interesting, I hope this research can expand to mammal and primates in the future.

Reviewer #3 :

See attached document.

Associate Editor Comments:

In this manuscript, author showed different recombination rates in different species and sex of mouse, and tried to use phylogenetic methods (tree construction and modeling) to study this difference and underlying evolutionary processes.

In general, results in this manuscript does show different mouse species and sex have different recombination rate (or MLH1 count), and this difference is related to phylogeny of mouse. Moreover, this manuscript prove that phylogenetic method can be a potential framework to further study sex-specific recombination rate.

Here I do have a few suggestions and questions which hopefully can improve this manuscript.

Method section

1. First, authors described mouse used in experiments, however, I noticed the ages of male and female mice are kind of different (line 127-128, 8-26 weeks v.s. 16.5-18.5 days post coitus), I was wondering will this difference in age influence recombination rate (or MLH1 count)?
2. In the part of phylogeny, I was wondering what software (such as Clustal Omega or MUSCLE) authors used for multiple sequence alignment (MSA)? Or is there any special step to construct phylogenetic tree was performed to replace MSA?
3. Authors tried to construct two trees for further robustness test, however, I was wondering why did authors use neighbor joining method (line 232)? I understand that NJ is very good under certain circumstances, however, in general I prefer to use probabilistic method (such as Bayesian) in addition to ML.
4. I personally appreciate that authors included control file of phymml, however, I think the code of control file should be put in data availability to save some space.
5. I think it will be helpful to mention what is "LD" (line 232) and talk a little bit more about AIC (line 277).
6. Line 253 and line 264 show two equations, I think it will be very helpful to add equation number after each equation, which will be easier to cite in other sections.

Result section

1. The first paragraph of result (line 285 to line 290) gives a description of MLH1, I think introduction or method section should be better place for it.
2. From line 382 to line 386, authors show the phylogenetic heritability on larger dataset, which has uneven number of male and female mice. In the next paragraph (line 388 to line 393), authors show result of smaller dataset with the same conclusion, I was wondering if this is still necessary to show larger dataset? When I read this part, I feel the experimental design is not very prefect because of different number of male and female, until I see the data from smaller dataset.
3. From line 412 to line 413, authors mention trees are "identical", here I assume authors mean the tree from the same method is the same, however, I misunderstood this as trees from different methods. I think maybe authors can say "ML (or NJ) trees informingare identical".

Discussion section

1. I do appreciate that authors pointed out some challenges in this research which is very inspiring. The only flaw is on line 572, I assume you tried to say "interlocus-mechanism" instead of "interlocus_mechanism".

In the study “Recombination Rates Are Governed by Sex-Specific Evolutionary Programs in House Mice,” the authors test the hypothesis that male and female mice have evolved under distinct evolutionary pressures. In addition to compiling MLH1 data from previous studies, they infer meiotic crossovers from MLH1 foci in adult testis and fetal ovarian cells in wild-derived strains. They construct their own phylogenetic tree based on LD-pruned SNPs and use a Brownian model of neutral evolution as a null hypothesis to predict the distribution of trait values across the sampled taxa. Their results reveal significantly higher phylogenetic heritability in male mice compared to female mice for this trait, suggesting natural selection selects for different trait optima between female and males. They point out that the fitness consequences of variation in recombination rate and genetic architecture underlying this sexual dimorphism are not yet well-understood. Overall, we found this manuscript to be well written with appropriate citations to the relevant literature (with one major exception noted below). Heterochiasmy is well-documented in the literature and yet understudied. A very recent study in birds documented differences in sex-specific genetic architecture, and this study elegantly shows differences in evolutionary trajectories between males and females in a hypothesis testing framework. Therefore, this study contributes important insights into this phenomenon and the potential for it to drive genetic conflict in mice.

Data availability: The supplement was easily accessible and downloadable. All data are publicly available or planning to be deposited into a public database.

Major criticisms:

- A recent study that is highly relevant is not cited here (likely because the authors were unaware) but the authors should read this article carefully and incorporate citations throughout where appropriate. The study (McAuley et al 2024. MBE, 41 (9) msae179) reveals differences in the genetic architecture between male and female house sparrows, which sets the stage nicely for the possibility of different evolutionary trajectories in the sexes.
- Unclear why they did not recalibrate base quality scores using GATK BQSR given how diverse the sequencing datasets were (different platforms, time points, etc.) Perhaps they can better justify in the methods how they dealt with the variation in datasets in choosing cutoffs for SNP calls.
- In Table 1, the means of each group are based on the mean of mean values across studies, but the SD is not based on the mean of SD values across studies, but instead is the SD of the means, which feels misleading. For example, while it is clear that the raw mean values for males have nearly twice the range as the females, the standard deviation within each study is ~2.5x higher in females than males, which does not seem to be a simple property of smaller

sample sizes. If anything, SD seems to increase with sample size (see plot below from values in Table 1). Instead, it would be better to either calculate mean/SD based on ALL the data instead of based on summary statistics. Or if summary statistics are used, they should be done consistently with the mean of mean values and the mean of SD values.

- Similarly, the increased standard deviation for females suggests that females have more variability within each study which is overlooked in the manuscript, where it is noted multiple times that females have reduced variance (lines 342-344; lines 426-427). This also suggests that the suggestion for more intense stabilizing selection in females on line 422 is unlikely and that perhaps the rapid evolution could be a possibility.
- Given the evidence that there seems to be selection favoring an increase in recombination rate, I think the reference to the Samuk et al 2020 study might be brought up a bit sooner (e.g. Para 2 of Discussion?). In this 2020 study, they attributed the increase in recombination rate to the differences in climate experienced by the two strains. Given that paragraph 2 of the discussion refers to geographic locations several times, perhaps a map of the distribution of these strains would be helpful to a broader audience. Such a figure would better help readers to understand if there are any potential environmental drivers that might explain the branches with elevated recombination rates. The general localities are referenced throughout the text, but again, a map figure might be more powerful in helping to see biogeographic patterns that perhaps explain (or do not explain) the phylogenetic signal.

Minor Criticisms and Typos:

- In Table 1, a legend would greatly improve the understanding. For example, the significance of the bolding was unclear at first.
- On lines 184-185, perhaps instead of percentage, the authors could add the specific difference in the counts of foci to make this point clearer. Squinting at the plots, they all appear within ~1 foci count, so that would perhaps be more useful

than saying 5.5%. This helps to confirm the lack of biological significance to the differences.

- On lines 546-550 in the discussion, the authors attempt to connect individual differences in fitness within sexes and recombination rate differences. This topic is discussed in this more recent modeling study (Drury et al 2023. GBE, 15(8): ead132) that had some very interesting insights that would be worth adding here.

Reviewer Responses

We thank the reviewers for their thoughtful and extensive critical feedback on our manuscript. We have addressed each of their comments in our point-by-point responses below and integrated their feedback into a substantially revised version of our manuscript, which we believe represents a marked improvement over the initial submission. Our responses are provided in black, with original reviewer comments emphasized in **bold blue**.

REVIEWER 1

In lines 80-83 (and indeed throughout the paper), the authors should acknowledge more that heterochiasmy is frequently underpinned by different genetic architectures in females and males. This has been repeatedly demonstrated in vertebrates (humans, cattle, pigs, deer, mice, sheep, goats, sparrows, salmonids) and often (but not always) implicates a suite of meiotic loci of modest to major effect to be involved (RNF212, RNF212B, PRDM9, SYCP2, MSH4, MEI1, etc). This is relevant for the current paper, as it provides a clear potential mechanism by which sex differences can rapidly evolve. It also provides a low-hanging fruit for future analyses of genomic data in this dataset by the authors to determine if there are phenotypic associations and/or signatures of selection.

We agree with the reviewer that the genetic control of heterochiasmy is an exciting and active research area. Our paper is specifically focused on the *evolutionary* mechanisms that give rise to sex differences in recombination rate, rather than genetic basis of these observations. Nonetheless, for a trait to evolve predictably, it must have a genetic basis, and we consider findings from this research field to be highly relevant to our work. We have added text to the introduction to emphasize prior work demonstrating sex-specific genetic control of mammalian recombination rates (page 4, lines 73-76).

While prior work has mapped loci with sex-limited or sex-specific effects on recombination rate in diverse vertebrate species (Kong et al. 2008; Kong et al. 2014; Liu et al. 2014; Ma et al. 2015; Halldorsson et al. 2019; Brekke et al. 2023; McAuley et al. 2024), the ultimate evolutionary causes of this sex dimorphism are largely unknown.

Our manuscript also features two paragraphs in Discussion that summarize the literature on sex-specific architecture of recombination rates, with particular emphasis on what is known about the genetic control of recombination rates in the house mouse system (pages 15-16, lines 557-583).

I had several questions about the phylogenetic modelling:

(a) It is not clear if the phenotypic value is a point estimate of the mean phenotypic value per strain, or if it incorporates all measures within each strain (possibly with the strain identity fit as a random effect to account for pseudoreplication).

(b) Given that the authors acknowledge potential differences between data sources, it may have been appropriate to fit the data source as a random effect in the model (to account for between dataset variance).

(c) To offer a potential solution if not possible in ape, I believe this is possible in MCMCgimm (Bayesian approach) that could allow better fitting and interpretation of the error around the estimates (which is never quite clear in the results of this manuscript).

Our original phylogenetic comparative analysis focused on fitting the mean MLH1 foci count for each sex and strain using the generalized linear mixed model approach proposed by Lynch in 1991 (PMID: 28564168). The reviewer's comments prompted us to deeply evaluate the

underlying assumptions, power, and rigor of this analysis. We realized that (1) the model's assumption of trait normality is not strictly appropriate, as MLH1 foci count data are bounded by the biological constraint for a minimum of one crossover per chromosome, (2) modelling mean MLH1 foci counts per strain barred us from estimating within strain variance, and (3) we could not explicitly account for study-level differences in this original modelling framework.

In the revised manuscript, we have reimplemented our analysis using Bayesian linear mixed-effect modeling (as implemented in the R package brms). This flexible modeling framework allows us to directly model other sources of possible variance as random effects, including effects due to differences between studies and between individuals within strains. Furthermore, the uncertainty in our estimates of phylogenetic heritability can be fully assessed by the posterior distributions for model variance components.

As we detail on pages 8-10 lines 243-318 of the revised manuscript, we separately fit male and female MLH1 counts to a Bayesian linear mixed-effects model, treating strain, individual nested within strain, and data source as random effects. The random strain effect is modelled as a multivariate normal distribution with variance structure shaped by the phylogenetic correlation matrix, thereby accounting for strain relatedness. In contrast to our original model, this new model fits MLH1 foci count observations from single cells as the response variable, rather than strain-level means. We more thoroughly describe our model fitting and evaluation approach on lines 285-303, as well as our approach for estimating phylogenetic heritability from estimated variance components (lines 305-318).

Importantly, we recover qualitatively identical results to our original findings using this new approach. The phylogenetic heritability (e.g., the proportion of variance explained by strain identity) is much higher for males ($H_p^2=0.81$; 95% HPD interval: 0.59-0.90) than females ($H_p^2=0.24$; 95% HPD interval: 0.05-0.54). Further, we show that the proportion of variance explained by study-level differences is modest (<16%; **Table 2**), confirming the findings shown in **Figure S2**. Similarly, inter-individual variation accounts for a small proportion of the variance in MLH1 foci counts (<10%; **Table 2**).

We have made significant text updates throughout the Methods and Results sections of the revised manuscript to reflect this change in our primary analysis method.

I must admit that I did not fully understand what the modelling of lineage-specific adaptive shifts was doing, or what it means biologically (Lines 270-277). It is not clear what "the inclusion of additional trait parameters" of an optimal trait and selection intensity really means, or what these are, or how these are fit (this indicates a reproducibility issue - it's not enough to just signpost the code). What does a unique trait optimum along a lineage mean? What are background lineages discussed in lines 449-461? I apologise if my comment is vague, but I still didn't fully understand what had been done after reading the results and discussion. Please help the reader more (even an apparently experienced one such as myself).

We have substantially revised the text of the relevant methods section to clarify the OU modeling (page 10, lines 320-337):

To explicitly model lineage-specific adaptive shifts in recombination rate, we used the Ornstein-Uhlenbeck (OU) phylogenetic modeling framework implemented in the R package ouch (Hansen 1997; Butler and King 2004; Cressler et al. 2015). OU models extend a neutral phylogenetic model of trait evolution (i.e., Brownian motion) by specifying the intensity of selection toward an optimal trait value along predefined lineages. Instead of trait values drifting randomly across a phylogeny, an OU model

*imagines traits being gently pulled toward a preferred state, much like a “magnetic” force. This magnetic pull is defined by three key parameters: the optimal trait value(s) favored by evolution (Θ), the strength of selection toward that optimum (α), and the amount of random variation around the optimum (σ). OU models provide flexibility to distinguish between distinct evolutionary models. Certain branches can be specified as “foreground” lineages, which are hypothesized to evolve toward one optimum, and the remaining “background” lineages, which evolve toward a second trait value. For both male and female recombination rates, we fit a model with a unique trait optimum along the *M. m. musculus* lineage compared to the rest of the phylogeny. This model was compared to two null models: strict Brownian motion and a model with a single trait optimum shared across all lineages. Model parameters were estimated via unconstrained numerical optimization to maximize the likelihood, with model selection performed through comparison of corrected AIC values.*

There is substantial discussion of sexual conflict in the discussion (lines 563-584, also in lines 49-50), but I think it should be toned down, as it is not directly supported by findings of this manuscript. First, I don't agree that differences in phenotype or allelic effects inherently mean there is conflict, e.g., that an allele that advances a trait value in one sex will always have a negative effect in the other sex (as implied in lines 565-567). Second, and more generally, I think that there is no evidence of high trait polymorphism, an arms race, variation in trait architecture etc. in this study that supports this argument. It would be good to know *how* these hypotheses could be tested in future work.

We respectfully disagree. First, we do not assert that sex differences in phenotypes or allelic effects imply genetic conflict. Instead, our assertion is that “sex differences in *trait fitness optima* impose an inherent genetic conflict”. Our original statement omits an important point: that genetic conflict can only emerge if the trait is influenced by the same loci in both sexes. Accordingly, we have rephrased the sentence on page 17 lines 617-619 in the revised manuscript:

Sex differences in trait fitness optima can impose a genetic conflict if alleles that advance the trait value in the direction favoring one sex have a negative effect on the fitness of the other sex.

We do not believe this to be a speculative or hypothetical statement, but rather a broadly recognized cause of sexual conflict (see, e.g., PMIDs: 29062057, 9533125, 16612887).

Second, we show and/or cite relevant prior work on page 17, lines 632-637 showing that recombination rates exhibit: “(1) high levels of recombination rate polymorphism within house mouse subspecies (Dumont and Payseur 2011b; Peterson and Payseur 2021), (2) unique, sex-specific patterns of accelerated recombination rate divergence across *Mus* (this study), and (3) the genetic control of recombination by X-linked and autosomal genetic loci with antagonistic effects in *M. musculus* (Murdoch et al. 2010; Dumont and Payseur 2011a)”. These observations (i.e., high levels of polymorphism, asymmetric rates of trait evolution between the sexes, and enrichment of antagonistic alleles on the sex chromosomes) are hallmarks of traits subject to genetic conflict (PMID: 23206133). Furthermore, we are careful to acknowledge that the genetic conflict hypothesis for recombination rate evolution is just that – a hypothesis (page 17, lines 637-641):

Thus, our findings layer new evidence onto the hypothesis that recombination rate variation across house mice may be driven, at least in part, by genetic conflict (Dumont 2017b). Additional work is clearly required to establish the validity of this hypothesis and

identify the aspects of meiosis that may be subject to conflicting evolutionary pressures in males and females.

There needs be more considerations on how differences in the female sample size may affect the estimates (e.g., what is observed when males are down-sampled to reflect a similar dataset), and how much of the signal could be driven by a few members of *Mus musculus* exhibiting particularly high male values.

We agree that the difference in sample size is a concern. Indeed, this recognition prompted us to down-sample our male dataset to match the number of strains for which we also had data for females (page 12, lines 429-433). As detailed in the manuscript and Table 2, down sampling the male data has no qualitative impact on our conclusions.

It is very likely that the signals we detect are driven by a few *M. musculus* males with exceptionally high values, and we acknowledge this possibility (page 15, 528-532):

*Curiously, elevated male recombination rates are observed in only a subset of our surveyed *M. m. musculus* strains (PWD/PhJ, PWK/PhJ, and GOR/TUA). Male recombination rates in other *M. m. musculus* strains (CZECHI/EiJ, CZECHII/EiJ, TOM/TUA, AST/TUA, and KAZ/TUA) are close, or even less than, the phylogeny-wide mean (**Figure 1**). Thus, the male-specific increase in recombination rate in *M. m. musculus* is not universal across this clade.*

We consider this outcome a key finding of our work and dedicate significant discussion to its interpretation in the Discussion (page 15, lines 532-555).

The abstract could benefit from having the actual values for the phylogenetic signal.

Agreed. We have added these values to the Abstract.

Lines 33-36: On my first read, I did not really understand what this text was saying. I think that it could be rephrased in terms of biological meaning - the female part is quite clear but it was less clear what a "unique adaptive optimum" in the male lineage means.

We have substantially revised the results portion of our Abstract for clarity. The relevant section now reads (lines 27-33):

*We show that the phylogenetic distribution of male recombination rates is well predicted by the underlying *Mus* phylogeny (phylogenetic heritability, $H_p^2=0.82$), contrasting with the weaker phylogenetic signal observed in females ($H_p^2=0.24$). *M. m. musculus* males exhibit a marked increase in recombination rate compared to males from other *M. musculus* subspecies, prompting us to test explicit models of lineage-specific evolution. We uncover evidence for an adaptive increase in male recombination rate along the *M. m. musculus* subspecies lineage, but find no support for a parallel increase in females.*

Line 172: Reproducibility: What scale is this quality score on, and how is this obtained?

The quality scores were assigned by Peterson and Payseur and are described in their paper. We have added a reference to our paper to refer the reader to the relevant parent manuscript (page 7 lines 177-179):

Within the Peterson and Payseur dataset, we further excluded cells assigned quality scores >4 to retain only the subset of high quality data (see (Peterson and Payseur 2021) for a description of quality scores).

Briefly, Peterson and Payseur assigned quality scores based “on visual appearance of staining and spread of bivalents”, with 1 assigned to the highest quality images and 5 to the worst. In their analysis, Peterson and Payseur exclude images with a score of 5; we adopt this practice in our work for consistency with their data analysis. MLH1 foci count data taken from other sources and newly collected for this analysis were not evaluated under a similar scoring strategy. We provide a detailed description of our procedure for obtaining high quality MLH1 counts from immunofluorescent images on page 6 lines 154-164.

General comment: Potential issues with karyotype variation are not discussed until quite late in the paper. A comment should be made earlier, and any species deviating from this should either be excluded or indicated in Tables and Figures.

With the exception of only the outgroup species, *Mus pahari*, all house mouse strains included in our analysis have a diploid karyotype comprised of 19 pairs of acrocentric autosomes and a pair of sex chromosomes ($2N=40$). Thus, karyotype is not a confounder in our analysis.

We have added a mention of karyotype conservation across *M. musculus* to the Introduction (page 5, lines 101-104):

These directional shifts in heterochiasmy and the magnitude of male recombination rate divergence in house mice are surprisingly stark given that M. musculus subspecies diverged only 350-500 thousand years ago (Phifer-Rixey et al. 2020) and share a conserved karyotype.

We also present this point in the second paragraph of the Results (page 11, lines, 357-362):

With the exception of M. pahari, which has a diploid chromosome number of 48, all Mus species in our dataset are represented by a conserved $2N=40$ karyotype comprised of acrocentric chromosomes. Thus, the absence of a clear species-level effect on recombination rate divergence is not simply a consequence of evolutionary shifts in the chromosomal constraints on recombination.

Table 1: What does the bold text represent?

The bold text presents group-level means. We have added the descriptors “Subspecies Average” and “Species Average” where appropriate in the strain column for clarification. We have also added footnotes to the table to describe how these group-level summary statistics were calculated.

Figure S1: a small point, but this visualisation does not represent the underlying data well (sample sizes etc.) - it would be better to have transparent boxes with the data points jittered underneath.

We appreciate these suggestions and have modified the figure to display all the data points. We have also updated the figure legend (now, Figure S2) accordingly.

Lines 181-186: I was a bit surprised that the authors did not deem differences in the inclusion of sex-chromosomes to be an issue, as this seems relevant to the study. See my point 2(b) above.

We agree that this is, on the surface, a surprising finding. However, as noted on page 6 lines 159-161, in house mice, “the meiotic dynamics of the XY sex chromosomes are temporally decoupled from those of the autosomes (Kauppi et al. 2011; Acquaviva et al. 2020).” As a result, the presence or absence of a MLH1 focus on the male sex chromosomes at pachytene is largely stochastic. Further, as the sex chromosome bivalent is only one of 20 in the cell, the

inclusion/exclusion of a focus on the sex chromosome necessarily represents <5% of the overall number of crossover events.

Data from Peterson and Payseur (2021) were the only data to include sex chromosome MLH1 foci. As shown in **Figure S2**, there is no consistent increase in MLH1 foci counts observed in their data. Indeed, Peterson and Payseur report lower MLH1 counts in the Gough Island stock, PWD, and SPRET compared to the current study and our prior work (Dumont et al. 2011b). We interpret these findings to suggest that strain drift over time, differences in housing environments experienced by mice in different studies, and/or technical differences between studies are equally, if not more, important contributors to variance. Further, as summarized above in response to point 2, our inclusion of study as a random factor in the revised Bayesian linear mixed-effect model indicates modest contribution of this variable to the overall variance captured by the model (~16%).

Lines 210-223: did the authors consider investigating potential effects of structural variants, or is this unlikely to be an issue?

We agree with the reviewer that structural variants play an important role in shaping the recombination landscape. However, SVs will only impact recombination when present in a heterozygous state. For example, heterozygous inversion carriers exhibit reduced recombination rates across the inverted region, often with compensatory increases in recombination elsewhere in the genome. With the exception of the outbred Gough Island mouse stock, our study focuses on inbred strains lacking allelic variation. Thus, we do not consider it likely that SVs meaningfully contribute to the patterns of strain variation we report.

Lines 225-243: what outgroups were selected, or is this not required?

We have added this important detail on page 8 line 238:

Both neighbor joining and ML trees were rooted to the M. pahari strain Mus pahari/EiJ...

Small comment: the paper reads like the results were supposed to be read before the methods, and also there is a bit of a Results & Discussion vibe in parts (e.g., lines 416-427). The paper was easy to read, so I don't mind it too much, but some information could have been useful earlier e.g., the karyotype information, divergence dates, known differences in mouse (consider whether Lines 312 - 324 could even be in the introduction).

We appreciate this perspective. We wrote the Methods section with the intent that it would be read before the Results. During revision, we have extensively re-written portions of the Methods and Results, keeping the reviewer's critique top-of-mind. This is manifest in the inclusion of significant text dedicated to establishing the theoretical context for our analyses (e.g., page 8, lines 263-265; page 10, lines 323-332). If there remain places where the reviewer believes information is missing or misplaced, we would be grateful to have them pointed out.

We do include several sentences dedicated to data interpretation in the Results section. This was a deliberate and strategic choice on our part, we believe this placement improves readability by contextualizing our findings as they are presented. Similarly, some critical background information is provided in the Results. Again, this was a deliberate editorial choice to aid reader comprehension and minimize redundancy.

Figure 1: This is a lovely figure but I have a few suggestions. The gridlines could match each branch rather than the species count as it would help direct matching of branch tips

with the data point. The standard errors cannot be seen on most points - is this because they are so small? Seems odd based on the table information, but if so, could be good to indicate this fact in the legend. It may also be good to indicate the species of SPRET, PANCEVO, CAROLI in the legend.

We appreciate these helpful suggestions. We have modified the gridlines to align with the tree and changed the figure to present strain means ± 1 standard deviation, rather than standard errors. Because the number of cells assayed for most strain-by-sex combinations is large, the standard errors associated with mean MLH1 foci count estimates were too small to render on our original plot. We have also modified the figure legend to indicate the species designations for strains SPRET, PANCEVO, and CAROLI.

Lines 342-344: how was the variance calculated - as the raw values or that estimated from the phylogenetic model? I also believe it is more appropriate to compare coefficients of variation rather than raw variance between different groups.

Here, the variance is calculated as the variance in the observed average strain-level MLH1 foci counts. We have added the word “observed” to this sentence to make it explicit that these estimates derive from the underlying raw data (page 11, lines 382-383).

The variance in average observed strain-level MLH1 foci counts for M. musculus females is smaller than the corresponding measure for males (1.73 versus 2.77)...

As referenced above, we have also added footnotes to Table 1 to clarify how group-level summary statistics were computed.

As we are interested in the absolute variability in MLH1 foci counts between the sexes, we do wish to emphasize variance here, not the coefficient of variation.

Lines 357-359: I agree with the word tentative here - I don't think there is enough evidence to back up this statement. Perhaps it is more appropriate to put this as a future hypothesis in the discussion.

We carefully acknowledge the tentative nature of this hypothesis and revisit the role of house mouse demographic history in shaping observed patterns of recombination rate variation in the second paragraph of the Discussion section (page 15, lines 532-552).

Line 377: "not genetically determined" - I found this statement a bit odd. I get that this is a possibility, but I think there is a lot of evidence that there is likely substantial genetic determination, not only through major effect loci (see point 1 above), but also through heritable variation in chromatin conformation & accessibility.

We agree with the reviewer that this outcome is highly unlikely in view of the established genetic control of recombination rate. We have rephrased this sentence to emphasize possible sex differences in environmental sensitivity of recombination rates, rather than genetic differences (page 12, lines 413-416):

In contrast, if MLH1 foci counts are subject to strong stabilizing selection, evolve exceptionally rapidly, or are highly sensitive to environmental differences, strain variation in this phenotype may not mirror strain ancestry.

Lines 380-386: in reporting these results, it would be appropriate to give more statistical parameters, such as the range and distribution of values obtained under the resampling regime.

Agreed. In the revised manuscript, we present the median phylogenetic heritability estimate from simulated draws from the posterior distribution and the 95% highest posterior density interval.

Lines 418-419: I would argue that having more stringent genetic controls could lead to lower heritability as there would be less tolerance to variation in the trait? Similarly, in most other vertebrate systems where individual measures have been used, males often show lower phenotypic variance and lower heritability within populations. I'm not saying that this has to be the case in your data, but I think that it is worth considering.

The phylogenetic heritability explains the proportion of variation in the trait that is attributable to the shared genetic ancestry of the samples. We agree with the reviewer that high phylogenetic heritability does not necessarily imply that a trait has a strong genetic basis. Such a signal could also be driven by strong stabilizing selection or indicate rigid physiological constraints on the phenotype. Conversely, a trait with a high genetic heritability could have low phylogenetic heritability if it evolves rapidly. We point out these alternative interpretations on lines 463-473. Overall, genetic heritability cannot be used to predict tolerance to trait variation, as a trait with high genetic heritability can have high plasticity.

The reviewer points out that many traits in other vertebrate systems show reduced phenotypic variance and lower heritability in males than females. Our data do not follow the first pattern of reduced phenotypic variance in males: the phenotypic variance of MLH1 foci counts is higher in males than in females. This finding aligns with conclusions from Peterson and Payseur 2021 and Dumont and Payseur 2011b. Our study was not designed to quantify genetic heritability, and we cannot draw conclusions about relative genetic heritability from our data.

Lines 477-479: There have been several studies that have done this between species that should probably be acknowledged, including Lenormand & Dutheil (2005, PLoS Genetics), Cooney et al (2021, Evolution), Kivioja & Rastas (2024, BioRxiv).

We deeply regret the omission of these important and highly relevant citations. We have added citations to recognize prior work investigating recombination rate variation in a phylogenetic framework (page 15, lines 516-517).

General comment: given the discussion of hybrid male sterility in paragraph 524-538, could PRDM9 play a role in heterochiasmy.

This is an interesting question. We consider it unlikely that PRDM9 plays a major role in heterochiasmy for two reasons. First, prior work has shown that male and female house mice with the same PRDM9 allele use the same set of PRDM9 hotspots, albeit with different frequencies and intensities (PMID: 30185906). Thus, there are notable sex differences in the fine-scale recombination landscape that must be due to factors other than PRDM9. Second, other studies have shown that PRDM9 has no to little effect on the overall genome-wide number of crossovers in mammals (e.g., PMIDs: 26840484, 24270358, 18239090). As MLH1 foci counts capture the latter dimension of recombination rate variation, it is unlikely that allelic variation in PRDM9 contributes heavily to our phenotype.

REVIEWER 2

First, authors described mouse used in experiments, however, I noticed the ages of male and female mice are kind of different (line 127-128, 8-26 weeks v.s. 16.5-18.5 days post coitus), I was wondering will this difference in age influence recombination rate (or MLH1 count)?

In mammals, female meiosis is initiated in the fetal gonad whereas males do not initiate meiosis until sexual maturity (~6-8 weeks in mice). While male and female mice used in our study are necessarily of different ages, we utilized meiocytes from an identical sub-stage of meiosis: pachytene.

In the part of phylogeny, I was wondering what software (such as Clustal Omega or MUSCLE) authors used for multiple sequence alignment (MSA)? Or is there any special step to construct phylogenetic tree was performed to replace MSA?

We obtained a variant call set by mapping each genome to the mouse reference sequence (described on page 7 lines 208-209). This yielded a set of SNPs on an identical genome coordinate system, obviating the need for multiple sequence alignment.

Authors tried to construct two trees for further robustness test, however, I was wondering why did authors use neighbor joining method (line 232)? I understand that NJ is very good under certain circumstances, however, in general I prefer to use probabilistic method (such as Bayesian) in addition to ML.

We share the reviewer's preference for probabilistic approaches to phylogeny construction. Our primary reason for using an NJ tree is that the NJ method affords fast genome-wide phylogenetic inference and requires minimal compute power. At the time of original analysis, we attempted to create a genome-wide Bayesian phylogeny from our 29 samples using MrBayes. Unfortunately, our efforts bumped up against the run time limits of our high-performance computing resources. As a work-around, we attempted to generate a Bayesian tree using only SNPs on chr19 (the smallest mouse chromosome). Approximately 50 hours of compute time were required to complete 500,000 iterations of the MCMC chain, with sampling performed every 1000 generations. Under these run settings, the estimated sample sizes (ESS) for most model parameters were significantly less than 100, indicating undersampling of the parameter space. We estimate that ~5-10 million MCMC iterations would be required to achieve adequate ESS (>100), which would still outstrip the job time constraints on our compute resources (maximum 2 week run time).

As we were unable to generate a robust Bayesian tree, we have opted to feature analyses based on NJ and ML trees in the paper.

I personally appreciate that authors included control file of phylml, however, I think the code of control file should be put in data availability to save some space.

We have moved this code to Supplemental File 1, which also includes the Newick format trees.

I think it will be helpful to mention what is "LD" (line 232) and talk a little bit more about AIC (line 277).

We have added a sentence on page 8 lines 222-226 to describe our rationale for performing the LD-thinning step and to explicitly define the acronym LD:

Briefly, variants were greedily thinned based on linkage disequilibrium (LD) to include only those sites with $r^2 > 0.2$ using PLINK (v2.00a2.3LM) (Purcell et al. 2007). This step reduced the VCF file from 152.85 million to 21.20 million SNPs and eliminated SNPs in perfect or high LD, maximizing the informativeness of remaining sites and reducing the computational burden of phylogenetic inference.

Similarly, we have defined the acronym AIC and added the following text on page 10 lines 335-338:

Model parameters were estimated via unconstrained numerical optimization to maximize the likelihood, with model selection performed through comparison of corrected Akaike Information Criterion (AICc) values. Specifically, the model with the lowest AICc was interpreted as providing the best fit to the data.

Line 253 and line 264 show two equations, I think it will be very helpful to add equation number after each equation, which will be easier to cite in other sections.

Agreed. We have added equation numbers in the revised manuscript.

The first paragraph of result (line 285 to line 290) gives a description of MLH1, I think introduction or method section should be better place for it.

We have moved this information to the Methods section entitled “Assaying autosomal crossover rate via cytogenetic analysis of MLH1 foci in pachytene cells”.

From line 382 to line 386, authors show the phylogenetic heritability on larger dataset, which has uneven number of male and female mice. In the next paragraph (line 388 to line 393), authors show result of smaller dataset with the same conclusion, I was wondering if this is still necessary to show larger dataset? When I read this part, I feel the experimental design is not very perfect because of different number of male and female, until I see the data from smaller dataset.

We acknowledge that our experimental design is imperfect, with MLH1 foci counts available for more males from more unique strains than females. This difference owes, in large part, to the technical challenge of collecting recombination rate estimates from female mammals, where recombination occurs in the fetal ovary. Timing matings in order to isolate female pups at the correct developmental stage is challenging for wild-derived inbred strains that (1) are often poor breeders and (2) rarely have externally visible mating plugs. Further, the dissection of microscopic neonatal ovaries requires incredible dexterity and practice.

While these challenges have minimized the number of strains represented in the female dataset, we do wish to present our phylogenetic heritability estimate on the full male dataset. Pruning the male dataset down to include to the subset of strains with female data necessarily omits a large amount of informative data. Nonetheless, as we show on page 12, lines 430-434 and in Table 2, thinning the male data has no impact on our qualitative findings or conclusions.

From line 412 to line 413, authors mention trees are “identical”, here I assume authors mean the tree from the same method is the same, however, I misunderstood this as trees

from different methods. I think maybe authors can say “ML (or NJ) trees informingare identical”.

We have made the suggested wording change.

The only flaw is on line 572, I assume you tried to say “interlocus-mechanism” instead of “interlocus_mechanism”.

We have corrected this typo.

REVIEWER 3

A recent study that is highly relevant is not cited here (likely because the authors were unaware) but the authors should read this article carefully and incorporate citations throughout where appropriate. The study (McAuley et al 2024. MBE, 41 (9) msae179) reveals differences in the genetic architecture between male and female house sparrows, which sets the stage nicely for the possibility of different evolutionary trajectories in the sexes.

Thank you for calling attention to this relevant recent paper. Several other studies have identified sex differences in the genetic control of recombination and are referenced on page 4 lines 74-75. We have included this additional citation.

Unclear why they did not recalibrate base quality scores using GATK BQSR given how diverse the sequencing datasets were (different platforms, time points, etc.) Perhaps they can better justify in the methods how they dealt with the variation in datasets in choosing cutoffs for SNP calls.

We performed variant calling using DeepVariant, not GATK. DeepVariant take as input aligned sequencing reads. It generates pileup image tensors from those reference-aligned reads, applies a convolutional neural network to classify each tensor and perform base and variant calling, and outputs variant calls in standardized formats (e.g., gVCF, VCF). DeepVariant's CNN incorporates quality score information as part of its feature set. Thus, DeepVariant inherently accounts for sequencing errors and biases, which negates the need for separate recalibration step.

We selected this variant caller as it is suitable for both short-read (Illumina) and long-read (PacBio HiFi) variant calling, allowing us to use a single caller for the diverse genome sequence datatypes used in our phylogenetic analysis. DeepVariant is the caller recommended by PacBio for SNP and indel calling from HiFi data, and has also been shown to outperform GATK in benchmark tests of short read datatypes (PMID: 30247488).

In Table 1, the means of each group are based on the mean of mean values across studies, but the SD is not based on the mean of SD values across studies, but instead is the SD of the means, which feels misleading. For example, while it is clear that the raw mean values for males have nearly twice the range as the females, the standard deviation within each study is ~2.5x higher in females than males, which does not seem to be a simple property of smaller sample sizes. If anything, SD seems to increase with sample size (see plot below from values in Table 1). Instead, it would be better to either calculate mean/SD based on ALL the data instead of based on summary statistics. Or if summary statistics are used, they should be done consistently with the mean of mean values and the mean of SD values.

We clarify the calculation of these group-level summary statistics in newly added footnotes to Table 1. We also identified and corrected an error in our original calculation of the *M. musculus*-wide standard deviation.

Group-level means were calculated by averaging strain-level means (e.g., *M. m. castaneus* subspecies mean = $(21.96 + 22.67 + 24.02 + 22.67)/4 = 22.84$). Group-level standard deviations (SD) were calculated by computing the SD across the strain-level means (e.g., *M. m. castaneus* subspecies SD = $\text{sqrt}(\text{var}(21.96, 22.67, 24.02, 22.67))$). This was a carefully considered

calculation, as we are interested in the variability in MLH1 counts across strains from a single sex and subspecies. In essence, we wish to treat each strain estimate of the average MLH1 foci count as a random draw from the (unobserved) subspecies distribution of recombination rates to estimate the variance of this subspecies distribution.

The reviewer instead suggests calculating the variance across strain-level variance estimates. This estimate would be confounded by differences in sample size and data sources, complicating its biological interpretation. Similarly, calculating the mean or SD based on all the data for each group would disproportionately weight the contributions from strains with large amounts of data.

Similarly, the increased standard deviation for females suggests that females have more variability within each study which is overlooked in the manuscript, where it is noted multiple times that females have reduced variance (lines 342-344; lines 426-427). This also suggests that the suggestion for more intense stabilizing selection in females on line 422 is unlikely and that perhaps the rapid evolution could be a possibility.

We respectfully disagree with the reviewer's proposed alternative interpretation. We find increased variance for MLH1 counts in females *within strains*, while variance *across* strains is reduced in females compared to males. This finding mirrors previously reported observations in mice (PMID: 19535547) and humans (PMID: 18239090), and potentially owes to sex differences in (1) the global distribution of recombination (e.g., PMID: 9718341), (2) meiotic chromatin organization (e.g., PMID: 16175513), (3) recombination hotspot number and usage (e.g., PMID: 30185906), (4) developmental timing and progression of meiosis, or (5) the genetic control of recombination.

We have edited lines 383-385 to clarify that we are specifically referencing the between-strain variance in average MLH1 counts:

The between-strain variance in average MLH1 counts for M. musculus females is smaller than the corresponding measure for males (1.73 versus 2.77)...

Similarly, we have edited the text on lines 473-474 to make it explicit that we are referencing variance among strains:

*We discount this final interpretation as unlikely given the lower between strain variance in MLH1 foci counts in females compared to males (**Table 1**).*

The observation of reduced variance in female MLH1 counts across strains is consistent with more intense stabilizing selection in this sex. We note that Peterson and Payseur 2021 (PMID: 33683358) reached an identical conclusion based on analysis of a subset of the data presented in our manuscript.

Given the evidence that there seems to be selection favoring an increase in recombination rate, I think the reference to the Samuk et al 2020 study might be brought up a bit sooner (e.g. Para 2 of Discussion?). In this 2020 study, they attributed the increase in recombination rate to the differences in climate experienced by the two strains. Given that paragraph 2 of the discussion refers to geographic locations several times, perhaps a map of the distribution of these strains would be helpful to a broader audience. Such a figure would better help readers to understand if there are any potential

environmental drivers that might explain the branches with elevated recombination rates. The general localities are referenced throughout the text, but again, a map figure might be more powerful in helping to see biogeographic patterns that perhaps explain (or do not explain) the phylogenetic signal.

We have included a map of the strain origins as a Supplemental Figure (Figure S1). We have also added a reference to the Samuk et al. paper earlier in the Discussion, alongside an acknowledgement that the extremely high recombination rates observed in PWD, PWK, and GOR could reflect environmental adaptation (page 15, lines 553-556):

*It is also possible that high recombination rates in these strains reflect parallel adaptation to shared environmental pressures. Indeed, prior work has revealed that temperature can drive adaptive differences in recombination rates in *Drosophila* (Samuk et al. 2020).*

In Table 1, a legend would greatly improve the understanding. For example, the significance of the bolding was unclear at first.

We have added footnotes to Table 1 to clarify how subspecies and species-level summary statistics were calculated.

On lines 184-185, perhaps instead of percentage, the authors could add the specific difference in the counts of foci to make this point clearer. Squinting at the plots, they all appear within ~1 foci count, so that would perhaps be more useful than saying 5.5%. This helps to confirm the lack of biological significance to the differences.

The maximum difference in MLH1 count between studies profiling the same strain is 1.2 foci. We have added this number on line 191. However, we also retain the percentage difference in MLH1 foci number, as we believe it provides a better interpretation of the scale of variability between studies. For example, a difference of 1.2 foci in a strain with ~20 MLH1 foci would suggest a larger study effect than an identical numerical difference in a strain with an average of ~30 MLH1 foci.

On lines 546-550 in the discussion, the authors attempt to connect individual differences in fitness within sexes and recombination rate differences. This topic is discussed in this more recent modeling study (Drury et al 2023. GBE, 15(8): ead132) that had some very interesting insights that would be worth adding here.

We thank the reviewer for bringing our attention to this highly relevant paper. We have integrated a sentence into the revised Discussion to highlight pertinent findings from this simulation study (page 16, lines 597-600):

A recent simulation study concluded that human recombination rates are constrained by fitness costs for excessively high and low recombination rates, and that the fitness costs associated with excessively high recombination rates may be higher in males than females (Drury et al. 2023).

November 2, 2025

RE: GENETICS-2025-308628

Dr. Beth L. Dumont
The Jackson Laboratory
Mammalian Genetics
600 Main Street
Bar Harbor, Maine 04609

Dear Dr. Dumont:

Congratulations, your manuscript titled "Sex-Specific Evolutionary Programs Shape Recombination Rate Evolution in House Mice" is accepted for publication in GENETICS! Many thanks for submitting your research to the journal.

The reviewers had a few suggestions for improving the manuscript that you may want to consider when preparing the final version. You can view their comments at the bottom of this email.

To Proceed to Publication:

1. Format your article according to GENETICS style: <https://academic.oup.com/genetics/pages/author-guidelines>
2. Ensure that you comply with data and community resource citation guidelines: <https://academic.oup.com/genetics/pages/author-guidelines#section-5-9-2>
3. Upload your final files at <https://genetics.msubmit.net>
4. Add oupsupport@scipris.com and genetics.oup@novatechset.com (or the domains @scipris.com and @novatechset.com) to your email program's "safe senders" list. You will be contacted by both at various points during the production process.

Notes:

- Your currently-accepted manuscript (unedited, as submitted, reviewed, and accepted) will be published at GENETICS and deposited into PubMed as an Advance Access article. Notify sourcefiles@thegsajournals.org before signing your license if you do not wish to publish your article via Advance Access.
- We invite you to submit an original color figure related to your paper for consideration as cover art. Please email your submission to the editorial office or upload it with your final files. You can submit a small-sized image for evaluation, and if selected, the final image must be a TIFF file 2513px wide by 3263px high (8.375 by 10.875 inches; resolution of 600ppi). Please avoid graphs and small type.
- After files are sent to Oxford University Press we use SciPris to manage article licensing and payment. If you do not have a SciPris account, you will receive an email from no-reply@scipris.com to sign up to use Oxford University Press' author portal. After logging in, follow the online instructions to sign your license and arrange any payment due.

If you have any questions or encounter any problems while uploading your accepted manuscript files, please email the editorial office at sourcefiles@thegsajournals.org.

Sincerely,

Thomas Lenormand
Associate Editor
GENETICS

Approved by:
Anthony Long
Senior Editor
GENETICS

Review comments (if applicable):

Reviewer #1 :

Thank you for addressing my comments in detail - the message of the paper is much clearer. I believe that this study and its findings make a very important contribution to the literature on recombination and heterochiasmy. I have some one small remaining comment:

Line 473: "[males have] higher genetic heritability" - do the authors mean this in the phylogenetic sense, or in the contemporary population sense (additive genetic variance within strains)? The phrase "more stringent genetic controls" would imply to me that the trait would have lower contemporary heritability, as any stringency would keep genetic variation low.

Reviewer #2 :

In the revised edition, the authors appropriately addressed my questions about the previous version.

Reviewer #3 :

The authors have done an excellent job responding to reviewer criticisms. The updated manuscript, which was already of high quality, is much improved with significant clarifications. One point worth making is that the citation suggested "McAuley et al 2024. MBE, 41 (9) msae179" was suggested to be cited throughout, not simply in the introduction. For example, this citation is relevant to the discussion on lines 565-575. Overall, this continues to be a nicely done study on an understudied topic that will be of broad interest to the readers of Genetics.